# Identification of biomarkers for glycaemic deterioration in type 2 diabetes

We identify biomarkers for disease progression in three type 2 diabetes cohorts encompassing 2,973 individuals across three molecular classes, metabolites, lipids and proteins. Homocitrulline, isoleucine and 2-aminoadipic acid, eight triacylglycerol species, and lowered sphingomyelin 42:2;2 levels are predictive of faster progression towards insulin requirement. Of ~1,300 proteins examined in two cohorts, levels of GDF15/MIC-1, IL-18Ra, CRELD1, NogoR, FAS, and ENPP7 are associated with faster progression, whilst SMAC/DIABLO, SPOCK1 and HEMK2 predict lower progression rates. In an external replication, proteins and lipids are associated with diabetes incidence and prevalence. NogoR/RTN4R injection improved glucose tolerance in high fat-fed male mice but impaired it in male db/db mice. High NogoR levels led to islet cell apoptosis, and IL-18R antagonised inflammatory IL-18 signalling towards nuclear factor kappa-B in vitro. This comprehensive, multi-disciplinary approach thus identifies biomarkers with potential prognostic utility, provides evidence for possible disease mechanisms, and identifies potential therapeutic avenues to slow diabetes progression.

Type 2 diabetes is a progressive multifactorial disease which presently affects >400 m worldwide, with numbers expected to increase to >700 m by 2045[1]. Biomarkers for the disease, which provide a deeper understanding of the disease process, are therefore eagerly sought. Importantly, their identification may improve prediction and personalized approaches to disease treatment[2].

Whilst many studies have examined associations between circulating biomarkers and incident disease[3,4], to date few studies have explored changes associated with glycaemic deterioration after the development of diabetes. Published studies (reviewed in[5]) have established that faster glycaemic deterioration is seen in those who are diagnosed younger, are more obese at diagnosis, have lower HDL, and higher HbA1c. A few studies have investigated genetic variants associated with more rapid progression with small and variable results[5], although a report on a Hong Kong Chinese population reported a replicated finding that a high polygenic risk score consisting of 123 T2D risk variants was associated with increased progression to insulin requirement[6]. To date, no studies that have adopted a multiomic approach to biomarker discovery, or reported systematically how metabolites of different classes impact on progression. Such

associations have the potential to be clinically useful in terms of prediction, as well as providing biological insights into the processes that drive glycaemic deterioration in T2D.

In a collaboration based around the EU Innovative Medicines Initiative-2 Risk Assessment and ProgreSsiOn of Diabetes (RHAPSODY) we have undertaken here to identify, in three large European cohorts, biomarkers of diabetes progression of three molecular classes: charged small molecules (metabolites), lipids and proteins. In this way, we identify species and, in the case of two of the identified proteins, provide evidence through functional studies in preclinical models for previously unidentified mechanisms of action in disease-relevant tissues.

## Results

### Cohort characteristics and modelling of glycaemic deterioration

Individuals from three cohorts, DCS, GoDARTS and ANDIS were included. In a subset, molecular characterization was performed of which characteristics are shown in Table S1. The characteristics across the cohorts were comparable (Table S1). Male subjects were more abundant in the cohorts (>55%), and the average age ranged from

✉ e-mail: lmthart@lumc.nl; E.Z.Pearson@dundee.ac.uk; g.rutter@imperial.ac.uk

61–67 years with a BMI of 30–32 kg/m². Glycated haemoglobin (HbA1c) levels were on average lowest in DCS (median 47.08 mmol/mol), followed by GoDARTS (55.54 mmol/mol) and ANDIS (60.06 mmol/mol). The time from diagnosis to sampling time ranged from 0 to 2.63 years. Three phenotypic models were explored in the included cohorts which showed concordance with BMI, use of glucose-lowering drugs being risk factors and age, HDL and C-peptide being protective (Table S2).

## Metabolites are associated with increased diabetes risk and progression

Out of the 19 small metabolites examined, five were associated with disease progression with nominal significance in the base model (age, sex, BMI adjusted, $P < 0.05$) in the meta-analysis of three cohorts (Fig. 1). These were homocitrulline (Hcit), aminoadipic acid (AADA), isoleucine (Ile), glycocholic acid (GCA), taurocholic acid (TCA). Out of the five, the association of two remained significant after multiple testing adjustment, including aminoadipic acid (AADA, HR = 1.11, 95% CI = 1.01–1.22, $p_{FDR} = 0.03$) and homocitrulline (Hcit, HR = 1.12, 95% CI = 1.00–1.25, $p_{FDR} = 0.04$, Fig. 1, Table S3). Of note, for AADA higher levels were observed at baseline for incident insulin users versus non-insulin users, but not for homocitrulline (Supplementary Fig. 2). Furthermore, homocitrulline showed a modest interaction with BMI ($P = 0.03$) which could, to some extent, mask the differences in levels at baseline. For AADA, however, an interaction with C-peptide was observed ($P = 0.01$). Both metabolites showed associations in the same direction in the replication cohorts, but non-significant with attenuated effect sizes (AADA, HR = 1.03, 95% CI = 0.96–1.11; Hcit HR = 1.03, 95% CI = 0.88–1.21). In external validation cohorts, Hcit showed a trend as a risk factor for incident diabetes (HR = 1.05, 95% CI = 0.74–1.48) in

MDC. Based on a logistic model in DESIR, Hcit was a risk factor for prevalent diabetes (OR = 1.32, 95% CI = 1.05–1.66), but not incident diabetes (HR = 0.97, 95% CI = 0.73–1.30). AADA has previously been associated with a higher risk of incident type 2 diabetes in Wang et al. (OR = 1.60, 95% CI = 1.19–2.16)[7]. Finally, the most consistent risk factor for time to insulin was isoleucine level, which was nominally significant in the discovery cohort (HR = 1.09, 95% CI = 0.96–1.25), a risk factor for incident diabetes in MDC (HR = 1.48, 95% CI = 1.26–1.74) and DESIR (OR = 23.88, 95% CI = 3.13–182.31) as well as prevalent diabetes (OR = 10.94, 95% CI = 3.94,30.32). In addition, isoleucine levels showed a modest interaction with BMI ($P = 0.02$). Finally, GCA and TCA were modest risk factors for time to insulin requirement, with hazard ratios of 1.09 (95% CI = 0.91–1.31) and 1.06 (95% CI 0.99–1.15), respectively. In the replication set both TCA and GCA were in the same direction, but no longer significant with hazard ratios of 1.09 (95% CI = 0.91–1.31) and 1.04 (95% CI = 0.94,1.12).

## Plasma triglyceride levels are markers of diabetes progression and incident diabetes

Among the 162 lipids investigated, the levels of nine reached significance in the base model (Fig. 2). Among these eight lipids were a risk factor for early insulin requirement, and these were all triglycerides (Fig. 2, Supplementary Data 1). These eight lipids were also a risk factor for incident diabetes in MDC (Fig. 2). A single lipid was protective for early insulin initiation (SM 42:2;2, HR = 0.85, 95% CI = 0.73–0.99). Interestingly, SM 42:2;2 was a risk factor for incident diabetes in MDC (HR = 1.16, 95% CI = 1.06–1.27). Further adjustment in the discovery cohort attenuated the effect size but the direction remained the same. Furthermore, in the partly (HDL, C-peptide) and

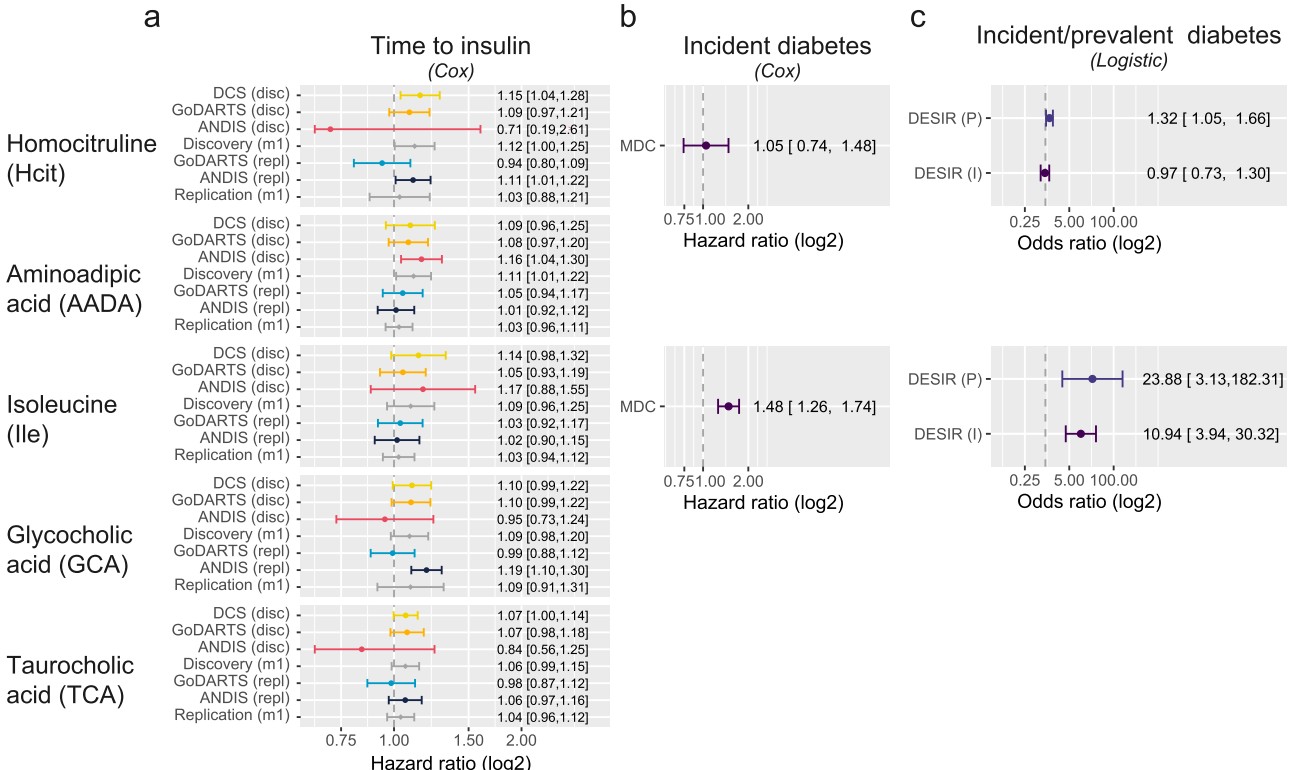

**Fig. 1 | Metabolites associated with diabetes development and progression.** **a** Hazards of a time to insulin model in the three discovery cohorts plus two replication sets in two of three discovery cohorts and their respective meta-analyses (Model 1). The figure shows the five nominally significant metabolites, with Hcit and AADA being also significant after multiple testing. Data are presented as hazard ratios with 95% confidence intervals. N = 1,267 individuals for DCS, n = 897 individuals for GoDARTS discovery, n = 699 individuals for GoDARTS validation,

n = 811 individuals for ANDIS discovery, n = 1969 individuals for ANDIS validation. **b** Hazards of incident diabetes in MDC based on a Cox proportional hazards model adjusted for age, sex, and BMI. Data are presented as hazard ratios with 95% confidence intervals. N = 3423 individuals **c** Odds ratios of incident and prevalent diabetes in DESIR based on a logistic regression model adjusted for age, sex and BMI. Data are presented as odds ratios with 95% confidence intervals. N = 1087 individuals for DESIR.

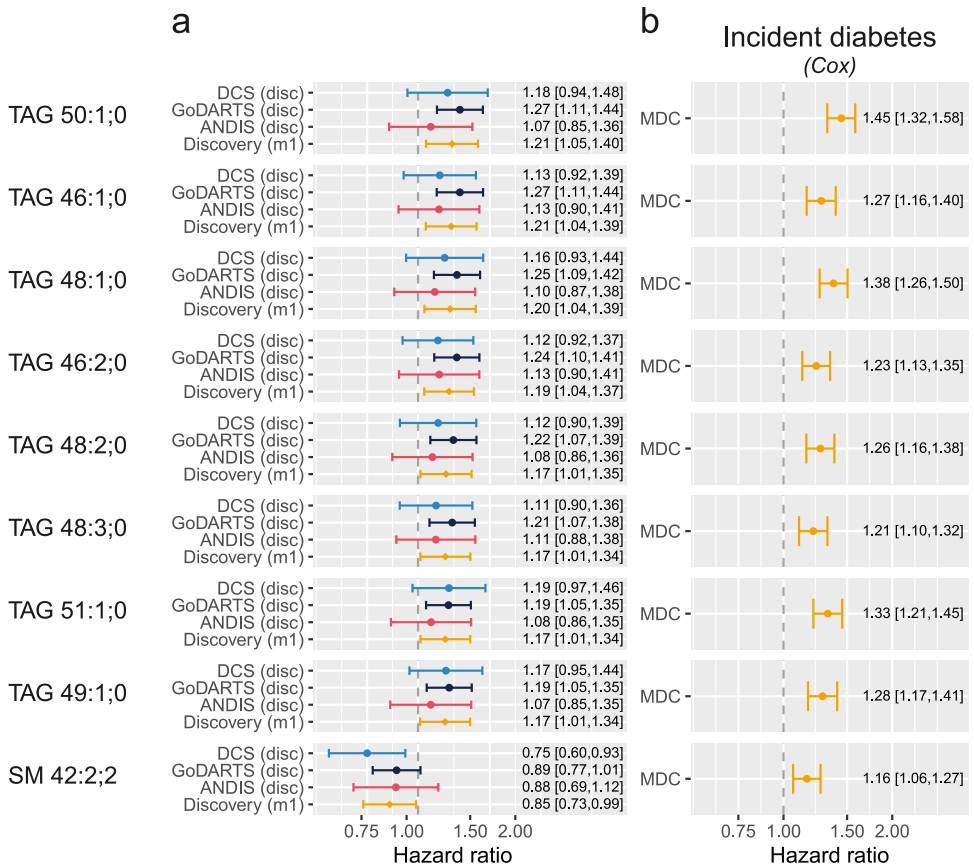

**Fig. 2 | Lipids associated with diabetes development and progression. a** Hazards of a time to insulin model in the three discovery cohorts and the meta-analysed hazards (Model 1). The figure shows the nine significant lipids after multiple testing. Data are presented as hazard ratios with 95% confidence intervals. *N* = 900 individuals for DCS, *n* = 899 individuals for GoDARTS, *n* = 809 for ANDIS. **b** Hazard models of incident diabetes in MDC based on a Cox proportional hazards model. Data are presented as hazard ratios with 95% confidence intervals. *N* = 3667 individuals.

fully adjusted model (additional adjustment for diabetes duration, glucose-lowering drugs) four and three lipids remained significant, respectively (Supplementary Data 1). At baseline, the levels of TAGs were higher in incident insulin users versus non-insulin users (Supplementary Fig. 3). In line with the protective hazard ratio, the levels of SM 42:2;2 were lower in the incident insulin users versus non-insulin users (Supplementary Fig. 3). As observed previously by us[8] based on a previous report[9], TAG acyl chain length and number of double bonds determined the magnitude of effect of TAGs. In this study, we also observe an almost linear relation between the acyl chain length and the number of double bonds and the hazard ratio, where the highest hazard was observed for the TAGs with the shortest acyl chains and the lowest number of double bonds (Supplementary Fig. 4). Nonetheless, the levels of TAGs were strongly correlated among each other (Supplementary Fig. 5).

## Plasma proteins levels associate with diabetes progression and prevalent and incident diabetes

In the 1195 investigated plasma proteins, the levels of 98 were nominally associated with time to insulin in the base model. Additional adjustment attenuated the hazard ratios only minimally in both the partly and fully adjusted model. MIC-1/GDF15 –from here onwards referred to as GDF15– was the protein associated with the highest risk of progression (HR = 1.34, 95% CI = 1.01–1.79) and this association was replicated in ACCELERATE (HR = 1.22, 95% CI = 1.04–1.42). Of note, GDF15 did not show a difference in baseline levels, but it should be noted that GDF15 levels are dependent on more factors including age (Supplementary Fig. 6). The protein associated with the second highest risk of progression was the Nogo receptor (NogoR, HR = 1.33, 95%

CI = 0.78–2.27, Fig. 3, Supplementary Data 2). In ANDIS, NogoR also replicated (HR = 1.20, 95% CI = 1.07–1.34, Fig. 3). NogoR was also a risk factor for incident (OR = 1.45, 95% CI = 1.15–1.83) and prevalent diabetes in AGES-Reykjavik (OR = 1.77, 95% CI = 1.60–1.95). In the top associated proteins, four were protective including SMAC, coactosin-like protein, testican-1 and HEMK2, of which HEMK2 was the most protective (HR = 0.78, 95% CI = 0.59–1.03). Levels of HEMK2 showed an interaction with C-peptide levels (*P* = 0.01). In the AGES-Reykjavik study, HEMK2 was also protective for prevalent diabetes (OR = 0.78, 95% CI = 0.72,0.85). Levels of testican-1 and HEMK, SMAC, coactosin-like protein were correlated (Supplementary Fig. 5). At baseline not all proteins showed a clear upregulation in level in incident insulin users; the most profound effects were observed for NogoR, IL-18 Ra, ENPP7, HSP 90b (Supplementary Fig. 6). Conversely, protein levels were downregulated particularly in Testican-1 and HEMK2 (Supplementary Fig. 6). Finally, aptamer specificity of the top proteins was verified by confirming the presence of a cis-QTL. Out of the eleven proteins, specificity of six proteins was verified on the basis of a cis-pQTL: GDF15, NogoR, IL18 Ra, CRELD1, ENPP7 and Fas (Table S4).

## Evidence of causality of biomarkers on incident diabetes based on Mendelian Randomisation

To assess causality of the identified biomarkers we would ideally have tested against the genetics of time to insulin requirement in people with diabetes, but in the current study the outcome was under-powered (n = 14,000) and there is no publicly available data for time to insulin genetic variants. Instead, we investigated the causality of biomarkers on type 2 diabetes. We found no significant associations with incident diabetes for any of the top metabolites (Table 1). However one

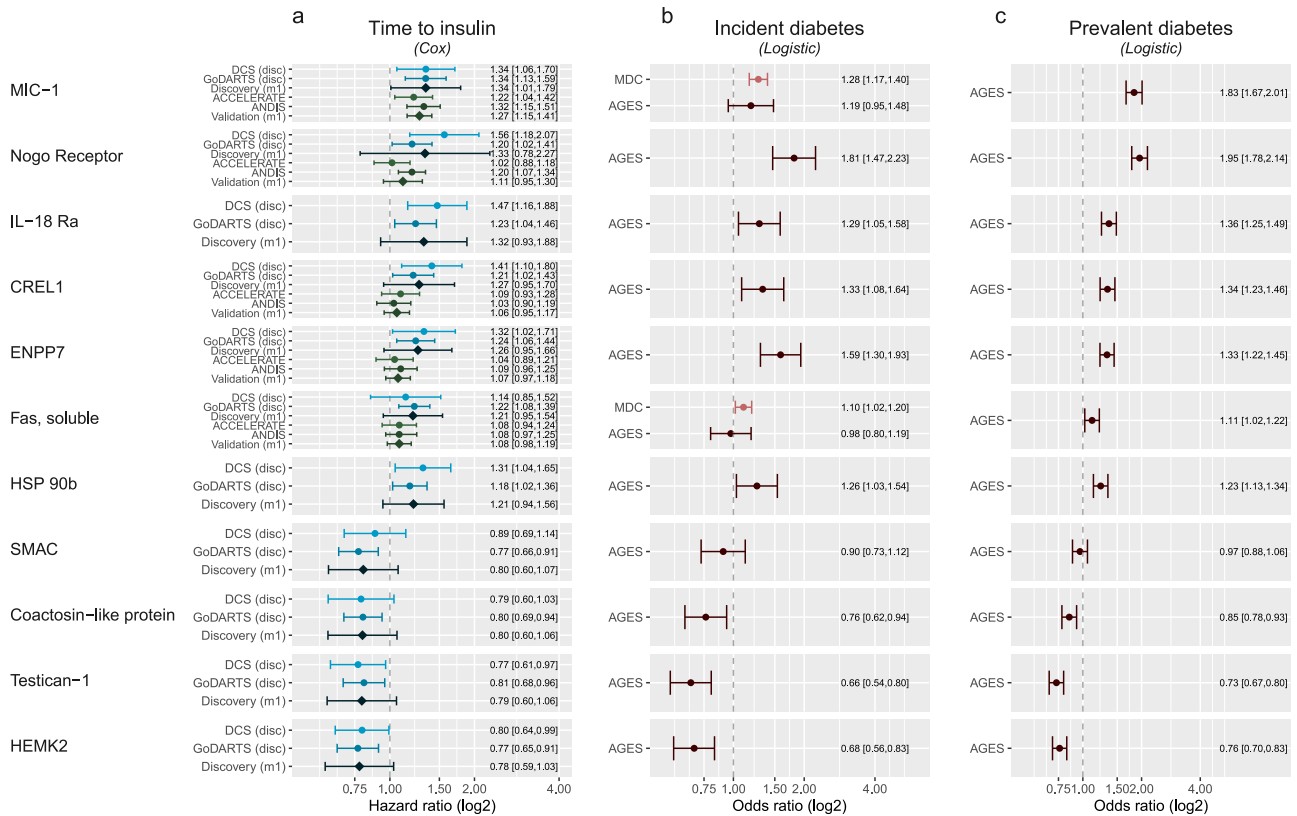

**Fig. 3 | Proteins in plasma or serum associated with time to insulin requirement. a** Top proteins associated with time to insulin requirement. Shown is the top 10 based on *P*-value plus Nogo receptor, which showed the largest risk of the top hundred proteins. *X*-axis, hazard ratio on a log2 scale and studies on the *y*-axis. Data are presented as hazard ratios with 95% confidence intervals. *N* = 589 individuals for DCS, *n* = 899 individuals for GoDARTS, *n* = 1992 individuals for ANDIS validation, *n* = 1850 individuals for ACCELERATE validation. **b** Association between protein levels and incident diabetes. *X*-axis, odds ratio on a log2 scale. Data are presented as odds ratios with 95% confidence intervals. *N* = 4915 individuals for MDC-CC and *n* = 5438 individuals for AGES. **c** Association between protein levels and prevalent diabetes. *X*-axis, odds ratio on a log2 scale. Data are presented as odds ratios with 95% confidence intervals. *N* = 5438 individuals for AGES.

## Table 1 | Mendelian randomization analysis on metabolites, lipids and proteins against incident type 2 diabetes

| **Metabolites** | | | | | | | | | |
| variable | method | nsnp | Beta | Lower | Upper | P-value | Egger intercept | Q | Q(df) | Q(P-value) |
|---|---|---|---|---|---|---|---|---|---|---|
| AADA | IVW | 2 | 0.02 | -0.09 | 0.13 | 0.77 | | 0.02 | 1 | 0.88 |
| Citrulline | IVW | 6 | -0.01 | -0.29 | 0.28 | 0.97 | -0.04 | 83.02 | 5 | 1.95·10⁻¹⁶ |
| Isoleucine | IVW | 5 | -0.06 | -0.94 | 0.83 | 0.90 | -0.11 | 127.69 | 4 | 1.22·10⁻²⁶ |
| Leucine | IVW | 6 | -0.08 | -0.74 | 0.58 | 0.81 | -0.07 | 127.77 | 5 | 7.09·10⁻²⁶ |
| **Lipids** | | | | | | | | | | |
| variable | method | nsnp | Beta | Lower | Upper | P-value | Egger intercept | Q | Q(df) | Q(P-value) |
| PE 18:0;0_18:2;0 | IVW | 3 | -0.07 | -0.12 | -0.02 | 3.89·10⁻³ | 0.02 | 5.53 | 2 | 0.06 |
| SM 42:2;2 | IVW | 2 | 0.00 | -0.07 | 0.07 | 0.99 | | 0.91 | 1 | 0.34 |
| TAG 50:1;0 | IVW | 2 | 0.01 | -0.04 | 0.05 | 0.75 | | 0.00 | 1 | 0.97 |
| **Proteins** | | | | | | | | | | |
| variable | method | nsnp | Beta | Lower | Upper | P-value | Egger intercept | Q | Q(df) | Q(P-value) |
| GDF15 | IVW | 8 | 0.03 | 0.01 | 0.05 | 2.68·10⁻³ | 0.01 | 8.83 | 7 | 0.27 |
| IL-18Ra | IVW | 14 | 0.02 | 0.003 | 0.03 | 0.01 | 0.00 | 6.75 | 13 | 0.91 |
| FAS | IVW | 4 | 0.05 | 0.005 | 0.09 | 0.03 | 0.00 | 6.34 | 3 | 0.10 |
| RTN4R | IVW | 10 | -0.01 | -0.03 | 0.01 | 0.28 | -0.01 | 4.16 | 9 | 0.90 |
| ENPP7 | IVW | 4 | 0.01 | -0.01 | 0.02 | 0.34 | 0.00 | 0.85 | 3 | 0.84 |
| CRELD1 | IVW | 7 | 0.01 | -0.01 | 0.02 | 0.43 | 0.01 | 7.33 | 6 | 0.29 |

*IVW* inverse variance weighting, nsnp, number of instruments. Statistical test MR: Inverse variance weighted regression; statistical test heterogeneity: Cochran's Q test.

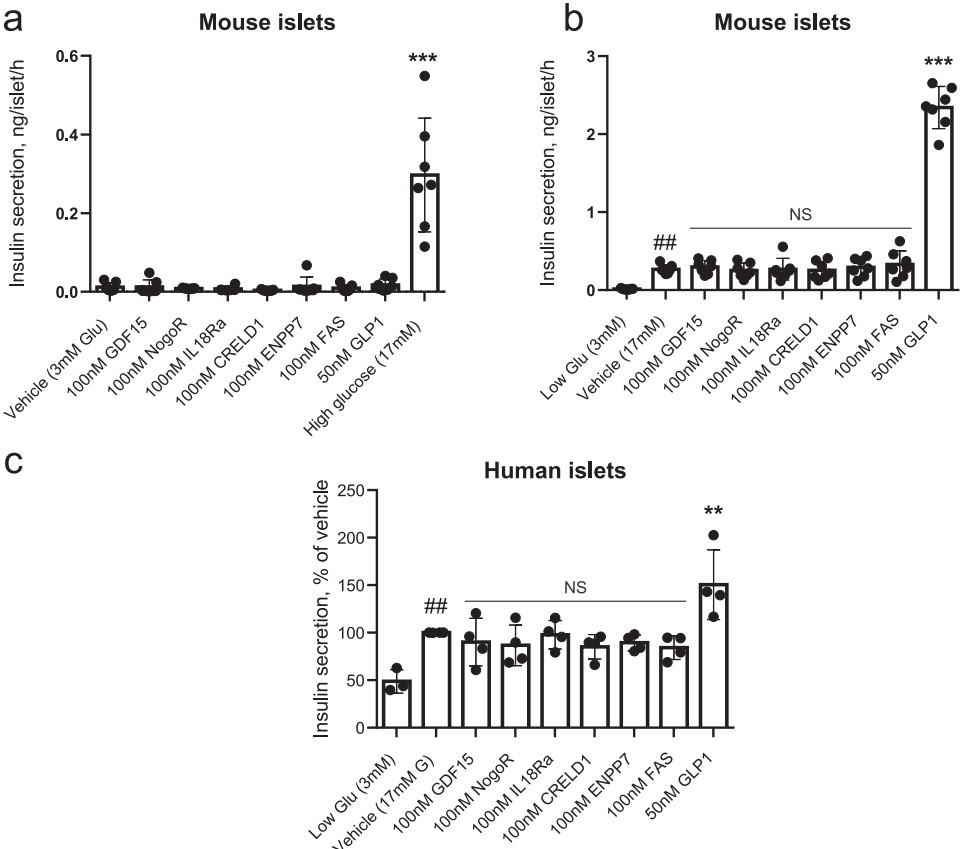

**Fig. 4 | Impact of identified biomarkers on insulin secretion from mouse (a, b) and human (c) islets.** Incubations were performed for 30 min. at the indicated concentrations of glucose, and secreted insulin measured using an electro-chemiluminescence assay. **a** ***$p = 3.19 \cdot 10^{-14}$ for the effects of 17 mM vs 3 mM glucose. **b** ##$p = 0.0074$ for the effects of 17 mM vs 3 mM glucose and ***$p < 2.2 \cdot 10^{-16}$ for the effects of 50 nM GLP-1 vs 17 mM glucose. **c** **$p = 0.0053$ for the effects of 50 nM GLP-1 vs 17 mM glucose and ##$p = 0.0097$ for the effects of 17 vs 3 mM glucose. Comparisons by one-way ANOVA in each case. Data points $n = 7$ replicates per treatment using islets from 16 mice (**a**, **b**) or those from individual human subjects ($n = 4$; **c**). Error bars represent means ± S.D. Other details are given in the Methods Section. Source data are provided as a Source Data file.

of the nominally significant phosphatidyl ethanaolamines, PE 18:0;0-18:2;0 has support for a causal association with type 2 diabetes (Table 1, $\beta = -0.07$, 95% CI = $-0.12$–$0.02$, $p = 3.89 \cdot 10^{-3}$). For the proteins, modest evidence of a causal relation was observer for three proteins, GDF15 ($\beta = 0.03$, 95% CI = 0.01–0.05, $p = 2.68 \cdot 10^{-3}$), IL-18Ra ($\beta = 0.02$, 95% CI = 0.003–0.03, $p = 0.014$) and FAS ($\beta = 0.05$, 95% CI = 0.005–0.09, $p = 0.03$). For the other protein biomarkers there was no evidence of a causal relation.

## Functional analyses of identified protein biomarkers

*Glucose-stimulated insulin secretion.* We chose to study six protein biomarkers with greatest effect size (GDF15, IL-18Ra, NogoR, CRELD1, FAS, ENPP7) and which accelerated progression, i.e., those which may plausibly exert a deleterious effect on insulin secretion or action. None of these affected basal (3 mM glucose) or high (17 mM) glucose-stimulated insulin secretion (GSIS) acutely (1 h) or after longer incubations (48 h) from either mouse (Fig. 4a, b) or human (Fig. 4c) islets. Glucagon-like peptide-1 (GLP-1) was used as a positive control, and stimulated secretion at the higher glucose concentration, as expected.

To determine whether any of the examined protein biomarkers might affect pancreatic beta cell mass, we next assessed their impact on beta cell apoptosis (Fig. 5a–c) and on proliferation (Fig. 5d) in mouse islets. Of those examined, only IL-18Ra (3-fold), and NogoR (>15-fold) exerted an effect, increasing apoptosis in mouse islets. Dose response analyses (Fig. 5b) revealed that the effects of NogoR were apparent only at high concentrations (>1 nM) likely to be above the

normal physiological range. At 100 nM NogoR, IL-18Ra also enhanced apoptosis in human islets (Fig. 5c). The increased apoptosis was further illustrated based on a TUNEL assay which measures apoptotic DNA fragmentation. A significant increase (Supplementary Fig. 7a–d, $P = 0.005$) in percentage of TUNEL positive nuclei−indicative of increased apoptosis−was observed in cells exposed to NogoR (8.5%) versus vehicle (3.7%). None of the tested compounds affected human islet proliferation (Fig. 5d).

## NogoR impacts glucose tolerance in vivo

GDF15 has been the subject of several earlier studies (see Discussion) and so was not pursued further here. Since NogoR was the biomarker with the next-largest correlation with disease progression, we next sought to determine whether this protein may influence glucose homoeostasis in vivo. Administered daily for 2 weeks to mice previously maintained on a HFHS diet (4 weeks[10]), we observed a clear improvement in glucose tolerance in vivo versus vehicle-injected animals (Fig. 6a–d). These changes were observed with no change in insulin secretion (Fig. 6e) or body weight (Fig. 6f). In contrast, when introduced chronically into the more severely diabetic db/db mouse model, NogoR had no discernible effect on oral glucose tolerance (Fig. 7a, b) or insulin secretion (Fig. 7c, f) but tended to increase fasting blood glucose and to impair insulin sensitivity (Fig. 7g, h), in line with the expected increase in circulating NogoR levels (Fig. 7i). Beta cell mass was not significantly affected by NogoR ($6.1 \pm 3.2$, $n = 5$ and $9.1 \pm 4.2$, $n = 4$, for NogoR-treated and control db/db mice, respectively, $p = 0.264$).

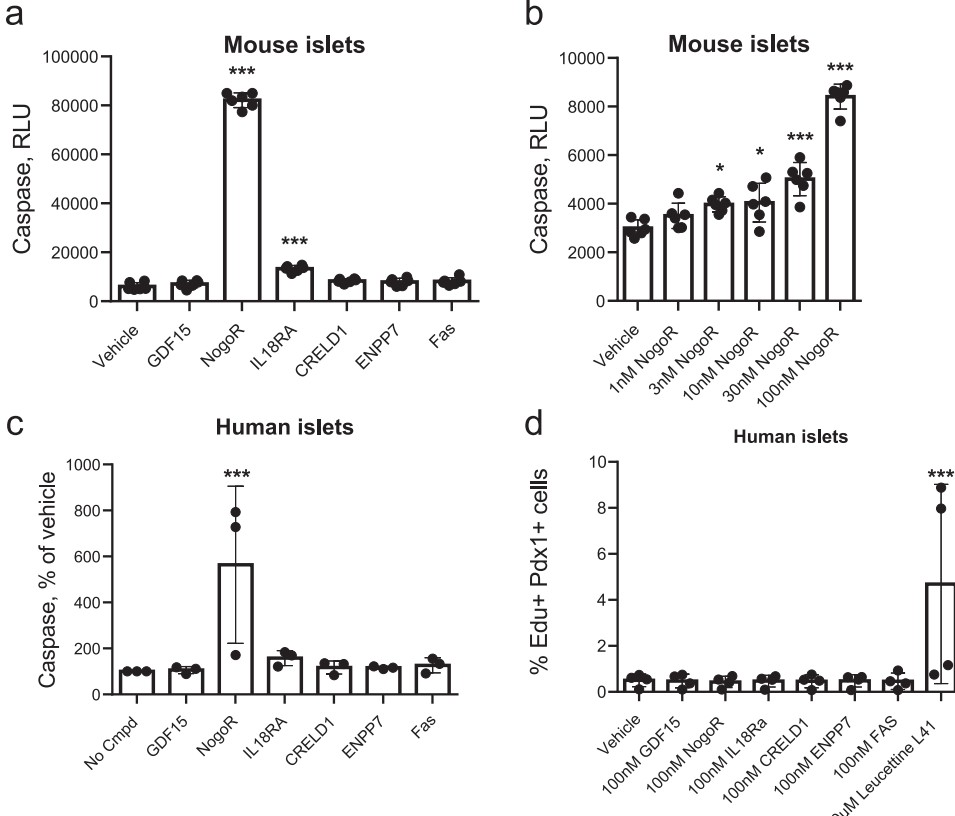

**Fig. 5 | Impact of identified biomarkers on apoptosis (a–c) ($n = 4$ replicates from four independent mouse islet preparations; 16 mice per preparation) or proliferation ($n = 4$ individual donors for human islets) (d) in mouse or human islets as indicated.** Test compounds were added at 100 nM unless otherwise indicated. **a** ***$p < 2.2 \cdot 10^{-16}$ for the effects of NogoR vs vehicle and ***$p = 5.45 \cdot 10^{-8}$ for the effects of IL18Ra vs vehicle; **b** *$p = 0.0196$, $0.0117$ for the effects of 3 nM and

10 nM NogoR, respectively *versus* vehicle by one-way ANOVA; ***$p = 1.67 \cdot 10^{-6}$ and $p < 2.2 \cdot 10^{-16}$ for the effects of 30 and 100 nM NogoR, respectively. **c** ***$p = 0.0009$; **d** ***$p = 0.0015$. The DYRK1A/DYRK2/CLK kinase inhibitor leucettine L41[86] was used as a positive control. Error bars represent means ± S.D. See Methods for other details. Source data are provided as a Source Data file.

To further explore whether NogoR might affect insulin signalling in disease-relevant tissues, we measured the action of this protein on signalling events downstream of insulin receptor activation in primary mouse hepatocytes (Supplementary Fig 8), in murine C3H10T1/2 adipocytes (Supplementary Fig. 9) and in human liver-derived HepG2 cells (Supplementary Fig. 10). Whilst we observed robust increases in insulin receptor beta subunit phosphorylation and in the phosphorylation of Akt on Ser473 and Thr308 in response to 100 nM insulin, no significant impact was observed on the acute responses to insulin over a range of different NogoR concentrations (1 nM, 10 nM and 100 nM) after treatment for 3–6 h with the biomarker in either mouse liver cells or pluripotent stem cell-derived adipocyte cells. (Supplementary Figs. 8, 9). Similarly, tested in HepG2 cells, no differences were observed in insulin-stimulated Akt phosphorylation in cells cultured over a range of CRELD1 concentrations (Supplementary Fig. 10).

### Effect of IL-18Ra on IL-18Ra signalling

After NogoR, IL-18Ra exerted the third strongest impact on diabetes progression (Fig. 3). We therefore tested the effects of IL-18Ra in a reporter cell line expressing the IL-18R and a luciferase construct under the control of the cytokine-regulated transcription factor, nuclear factor κB (NF-κB; Methods). IL-18Ra attenuated the actions of IL-18 over a range of concentrations at concentrations as low as 0.1 nM (Fig. 8).

## Discussion

We have undertaken a large multi-omic study, across three patient cohorts to discover lipid, metabolite and protein biomarkers for

diabetes progression. Many of our findings are replicated in independent diabetes progression cohorts or validated for incident and/or prevalent diabetes. In particular, we identify nine lipids, three small charged molecules and eleven protein biomarkers associated with accelerated glycaemic deterioration, and provide biological data in pre-clinical models demonstrating possible mechanisms of action for NogoR and IL-18Ra. Strikingly, measurements of proteins and lipid data reveal that the drivers of diabetes incidence and prevalence may be similar to those of progression.

### Metabolites and glycaemic deterioration

Two metabolites, Aminoadipic Acid (AADA) and Homocitrulline (Hcit) were significantly associated with diabetes progression, after correcting for multiple testing; three metabolites, Isoleucine (Ile), and the bile acids GCA and TCA, were nominally associated with progression. Isoleucine is a branched chain amino acid (BCAA) and a well-established risk factor for insulin resistance and increased risk of type 2 diabetes[11]. Correspondingly, Ile was also associated with both prevalent and incident diabetes in the current study. GCA and TCA are both existing markers for pre-existing diabetes[12].

In previous studies, AADA was shown to be a risk factor for incident diabetes[7] and Hcit with prevalent diabetes[13], consistent with our findings. AADA is an alpha amino acid formed as a downstream product of lysine oxidation by the action of myeloperoxidase (MPO)[14]. Higher levels of plasma AADA have been associated with obesity, insulin resistance, and increased risk of diabetes[7,15–17]. Hcit levels have been associated with disrupted energy metabolism in rat brains and have been linked to chronic renal failure[18,19]. Of note, however, where

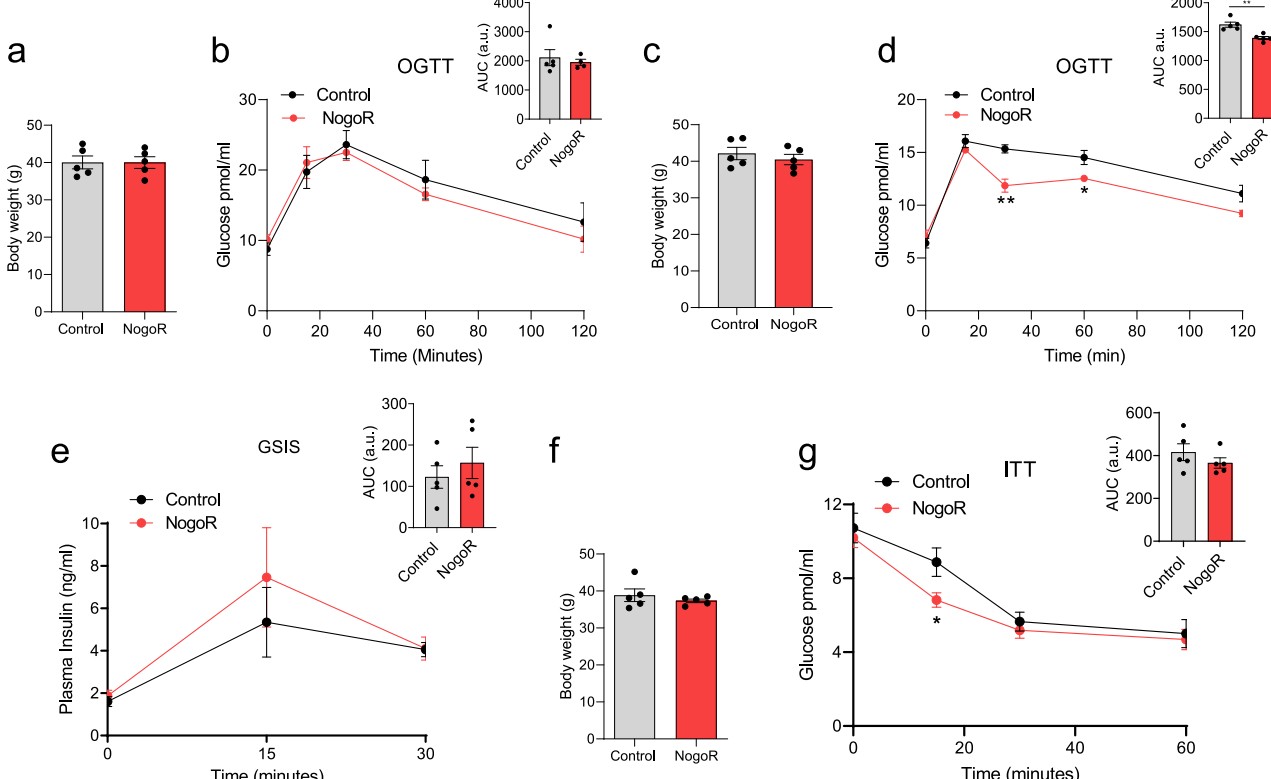

**Fig. 6 | NogoR enhances glucose clearance and insulin sensitivity in HFD mice.**
Two separate cohorts of wild-type male C57BL/6 J mice were maintained on a high-fat diet for 6 weeks, then injected for 14 consecutive days with saline or 100 ng (2.1 pmol/animal) recombinant NogoR. **a, b** Body weights of cohort one and circulating glucose levels during an oral glucose tolerance test (OGTT;2 g/kg) pre-NogoR treatment $n = 5$. **c, d** Body weights and blood glucose levels after an oral glucose load (2 g/kg) of cohort one after NogoR treatment. $n = 5$. **d** **$p = 0.0013$; *$p = 0.023$ by multiple unpaired $t$-test. AUC **$p = 0.0021$ by two-tailed unpaired Student's $t$-test. **e** Plasma insulin levels after an oral glucose load (2 g/kg) in cohort 1. $n = 5$ per group. **f, g** Post-treatment body weights of cohort 2 ($n = 5$ mice) and circulating glucose levels after receiving an intraperitoneal injection of 1 IU/kg of insulin. (i) *$p = 0.0462$ by multiple unpaired Student's $t$-test. Data are mean ± SEM. Source data are provided as a Source Data file.

there was a sufficient genetic instrument, none of the metabolites we assessed with mendelian randomisation were causally associated with diabetes risk, although it should be noted that BCAAs have been suggested to be causal in the aetiology of type 2 diabetes[20].

## Lipids and glycaemic deterioration
Nine lipids were associated with diabetes progression of which eight were associated with increased risk which all belonged to the Triacylglycerol (TAG) class. In the external validation data, all eight TAGs were associated with increased risk of type 2 diabetes. TAGs have previously been associated with incident T2D risk[21]. TAG species levels also strongly decline when individuals with obesity and diabetes undergo Roux-en-Y gastric bypass[22]. SM 42:2;2 was the sole lipid associated with lower risk on progression towards to insulin, but was associated with increased risk on future diabetes in the external validation data. A possible explanation for this could be that metformin treatment influences the sphingomyelin levels including SM 42:2;2 as has been shown in two studies in metformin treated HFD animals and human hepatocytes respectively[23,24]. The most-strongly associated lipids were also investigated for causality but generally instruments were not available or no evidence was observed. The MR analysis of PE 18:0;0_18:2;0 supported a possible causal relation with incident diabetes.

## Proteins and glycaemic deterioration
The protein with the strongest association with time to insulin was GDF-15. This protein has previously been implicated in diabetes incidence and the control of food intake[25,26], acting via receptors in the hind brain[10]. GDF-15 has also been reported to serve as a useful biomarker for impaired fasting glucose[27] and diabetic kidney disease[28] as well as a number of conditions including cardiovascular disease[29,30] GDF15 is strongly elevated following metformin exposure, due to release from the gut[31]. Of note, we did not show any attenuation of the GDF15 signal with progression when adjusting for whether the patients were treated with metformin at the time of blood sampling.

In addition, we identify several proteins biomarkers to be associated with glycaemic deterioration in diabetes, including NogoR (RTN4), IL-18Ra, CRELD1, ENPP7 and FAS. Interestingly, NogoR has previously been associated with prevalent diabetes and diabetes incidence[32]. In an effort to understand the potential impact of an increase in circulating NogoR on glucose metabolism, we demonstrated that injection of this biomarker improves glucose tolerance in high fat- high sucrose-fed mice, an effect likely to reflect improved insulin sensitivity; insulin secretion was not significantly affected (indeed tended to be increased after treatment). In contrast, when chronically infused into the severely hyperglycemic db/db mouse model, NogoR tended to reduce insulin sensitivity and further increase fasting glucose (Fig. 6). Thus, the effects of NogoR glucose metabolism in the whole animal are complex, and dependent on disease state.

NogoR, (encoded by the *RTN4R* gene), is chiefly expressed in the central nervous exist and mediates interactions between the myelin sheath and neurons, and has been implicated in Alzheimer's disease[33]. In this setting, NogoR interacts with Oligodendrocyte myelin glycoprotein (OMGP) and Nogo-A, present on myelin cells, to inhibit neuronal regeneration[34]. Thus NogoR serves as a receptor for Nogo-A but

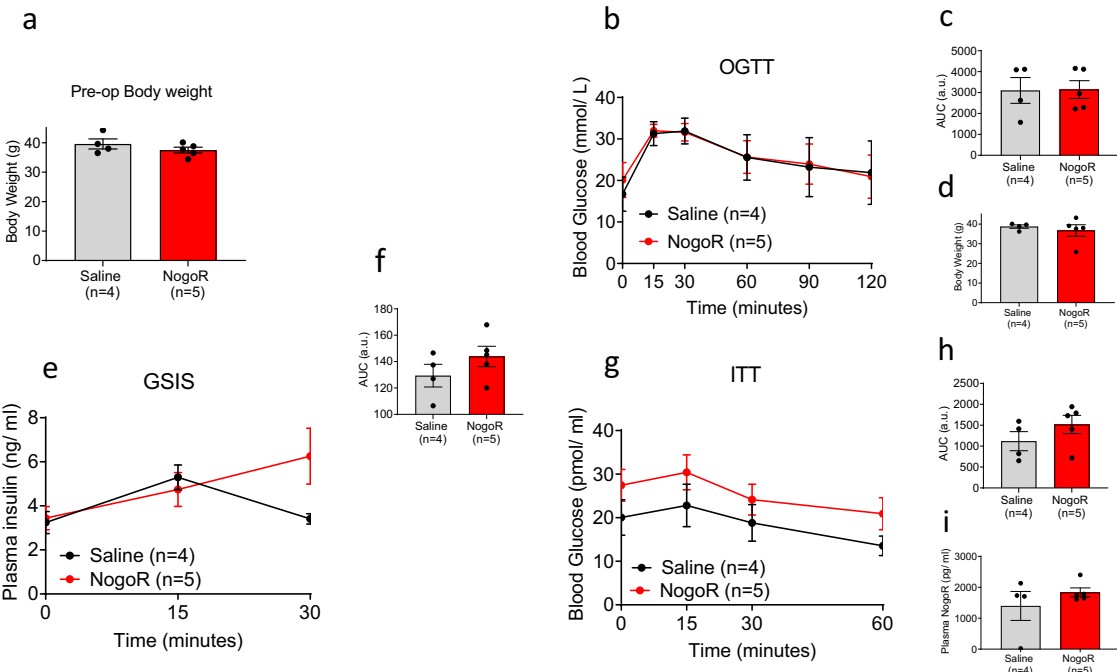

**Fig. 7 | NogoR has marginal effects on glucose clearance and insulin sensitivity in *db/db* mice. a** Body weights of *db/db* cohort pre-NogoR treatment *n* = 4–5. **b**–**d** Body weights, blood glucose levels and Area Under the Curve (AUC) after an oral glucose load (2 g/kg) 4 weeks after continuous NogoR treatment. *n* = 4–5. **e, f** Corresponding plasma insulin levels and AUC after oral glucose load (2 g/kg) shown in **b**. *n* = 4–5 per group. **g, h** Circulating glucose levels and corresponding AUC after receiving an intraperitoneal injection of 1 IU/kg of insulin. **i** Circulating NogoR levels 4 weeks after continuous treatment. Data are mean ± SEM. Source data are provided as a Source Data file. Data in **b**, **e**, and **g** were analyzed by multiple unpaired Student's *t*-test, and those in **a**, **c**, **d**, **f**, **h**, **i** by Mann–Whitney test. No significant statistical differences were detected.

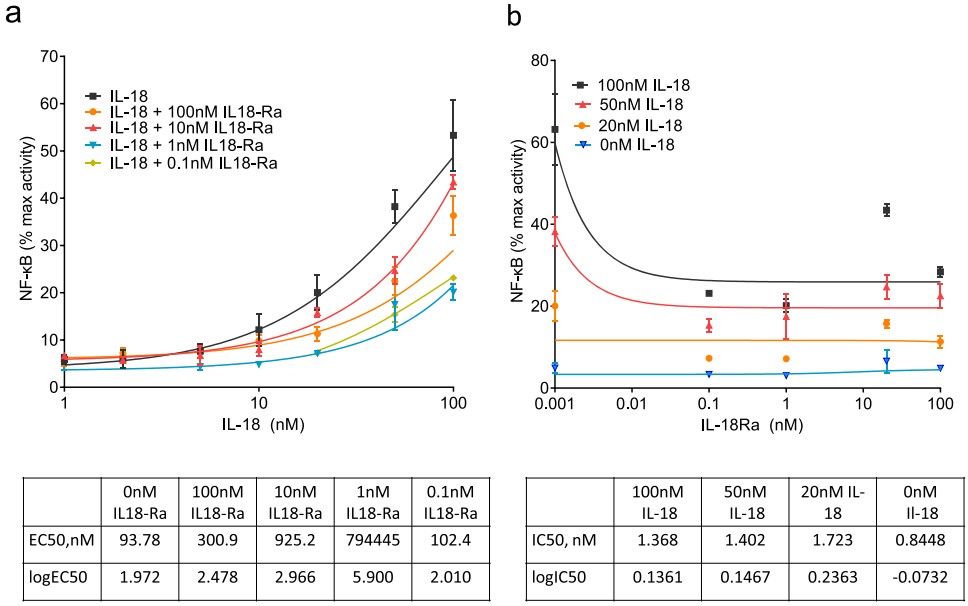

| | 0nM IL18-Ra | 100nM IL18-Ra | 10nM IL18-Ra | 1nM IL18-Ra | 0.1nM IL18-Ra |
|---|---|---|---|---|---|
| EC50,nM | 93.78 | 300.9 | 925.2 | 794445 | 102.4 |
| logEC50 | 1.972 | 2.478 | 2.966 | 5.900 | 2.010 |

| | 100nM IL-18 | 50nM IL-18 | 20nM IL-18 | 0nM Il-18 |
|---|---|---|---|---|
| IC50, nM | 1.368 | 1.402 | 1.723 | 0.8448 |
| logIC50 | 0.1361 | 0.1467 | 0.2363 | -0.0732 |

**Fig. 8 | NF-κB activation in HEK293 cells overexpressing IL-18R and co-stimulated with Il-18 and IL-18Rα.** IL-18R HEK cells stimulated for 6 h with 100, 50, 20, 10, 5, 2 and 1 nM of IL-18 alone, and also with 100, 10, 1 and 0.1 nM of IL-18Rα. IL-18 and IL-18Rα concentrations were tested in triplicate and NF-κB activation was measured using dual luciferase reporter assay. Data (means ± S.E.M.) are from three fully independent experiments. Curve fitting was done using non-linear regression with GraphPad Prism 9.0.0 for log(agonist) vs. response (**a**) and inhibitor vs. response (**b**). Source data are provided as a Source Data file.

conceivably also the shorter homologues Nogo-B and Nogo-C[35]. Importantly in the context of systemic glucose homoeostasis, Nogo-B interacts with the Nogo-B receptor (NgBR, encoded by *NUS1*) and knockout of *Nus1* in mice causes hepatic steatosis, possibly be interfering with insulin signalling[36]. Furthermore, variants in the human *NUS1* gene including rs4443534 are associated with altered type 2 diabetes risk[37]. Thus, by titrating Nogo-B (or other Nogo family members) circulating NogoR may act indirectly on the liver to affect

glucose storage or glycogen breakdown. On the other hand, NogoR also enhanced cell death in pancreatic islets, at least at high concentrations of this biomarker. The mechanisms involved in the latter action are unclear given very low levels of expression of the receptors NogoR (cell attached), p75 and OMGP in islets[38] (BioGPS.org) though the disruption of an interaction between the NogoR/p75 complex and the co-receptor Lingo-1[39], also expressed in islets, by soluble NogoR, is conceivable. Since *RTN4*, encoding NogoA, NogoB and NogoC via alternative splicing[40], as well as NgBR, are also expressed in islet endocrine cells (https://huisinglab.com/data/index.html), circulating NogoR might also disrupt NogoB binding to NgBR on $\beta$ cells.

Whilst the median concentrations of NogoR in the ACCELERATE cohort were ~2 ng/l (50 pM), a small number of individuals (70/1849) were found to display much higher levels (≥5 ng/L up to 90 ng/L or ~2.2 nM), concentrations close to those which elicited cell death in islets (Fig. 5). Nevertheless, we did not see—possibly due to limited power— a direct relationship between risk and having a very high NogoR plasma level, arguing against direct (patho)physiogical effects of these high levels in humans.

Moreover, we did not observe actions of NogoR on insulin receptor-proximal signalling events (phosphorylation of the receptor and of the protein kinase AKT) in three different cellular models of disease-relevant tissues (Supplementary Figs. 8–10). Further studies will be needed to explore more distal and physiological responses in each cell type (e.g. glycogen synthesis or glucose output in hepatocytes or triglyceride synthesis in adipocytes) as well as to explore the effects of NogoR at physiological levels of the NgBR receptor agonist NogoB.

CRELD1 (Cysteine-Rich with EGF-Like Domains 1, AVSD2, Cirrin) is a membrane-bound $Ca^{2+}$-binding member of the EGF family critically required for normal development of the heart[41] whose expression was recently shown to influence T-cell activity and immune homoeostasis[42]. We were, however, not able to test the effects of this agent on glucose homoeostasis in vivo due to the unavailability of the human protein, nor did our in vitro studies provide evidence that CRELD1 may regulate insulin signalling in vitro.

For IL18Ra it is unclear how its increased plasma levels contribute positively to the disease progression process. A recent Mendelian randomisation study[32] provided nominally significant support for a causal association between IL-18Ra and incident T2D, which was repeated here. Consistent with a role in glucose homoeostasis, IL-18 deletion in the mouse leads to obesity and insulin resistance[43]. Conversely, after weight loss following exercise/diet or bariatric surgery in man, a significant reduction in IL-18 concentrations was observed[44,45]. IL-18 secretion also increased in response to inflammasome activation and pyroptosis[46]. We confirmed earlier studies[47] which demonstrated that an IL-18Ra:Fc fusion fragment inhibits the pro-inflammatory action of IL-18. However, and in distinction to these earlier studies[47], measuring interferon-gamma (IFN-gamma) production from mononuclear cells, the actions of IL-18Ra, as examined in the present study, did not depend upon the additional presence of IL-18Rbeta. Other data suggest that IL-18Ra might interact directly with single immunoglobulin-IL-1-related receptor (SIGIRR, also called IL-1R8 and TIR8) which in turn inhibits anti-inflammatory signalling by IL-37[48]. We also noted that the concentrations of IL-18Ra as low as 0.1 nM efficiently inhibited the actions of a considerable excess (100 nM) of IL-18 potentially suggesting a non-competitive action (i.e., binding to the IL-18R at a separate site on the cellular receptor to IL-18). Unexpectedly, IL-18Ra also influenced beta cell apoptosis in vitro. In a previous study it was shown that IL-18 leads to the production of nitric oxide and apoptosis in mouse islets, while deletion of the IL-18 receptor accelerated graft failure, demonstrating an essential role for signalling by IL-18R to maintain islet viability[49]. The mechanisms involved here are unclear, however, since IL-18Ra expression levels in the beta cell are low[50].

Finally, we note that IL-18Ra, CRELD1 and coactosin-like protein are all involved in immune regulation, and might thus influence the inflammatory changes known to be involved in T2D[51]. Taken together, these findings suggest that sequestration of IL-18 by IL-18Ra, as well as direct signalling by IL-18Ra, may affect $\beta$ cell function and/or survival, and hence disease progression. Nevertheless, the contrasting actions of IL-18Ra (impaired IL-18 signalling in the HEK cell model of receptor overexpression, but increased apoptosis in primary islets) suggest that further studies, possibly involving tissues-specific inactivation of IL-18Ra or other potential receptors (see above), will be needed to fully elucidate the (patho)physiological roles of circulating IL-18Ra.

ENPP7 (Ectonucleotide pyrophosphatase/phosphodiesterase-7) is strongly expressed in the small intestine where it is involved in sphingomyelin hydrolysis and the absorption of ceramide and phosphocholine[52]. These processes might therefore be influenced by changed circulating levels of ENPP7. FAS (FAS cell surface death receptor; TNF superfamily receptor-6) is involved in caspase 3 and 8 activation and cell death[53] and might thus influence cell survival in critical metabolic tissues. HSP 90B (heat shock protein 90 alpha family class B member 1), encoded by *HSP9OAB1*, is a molecular chaperone and regulator of protein folding[54].

Other identified proteins were associated with protection from glycaemic deterioration of diabetes. SMAC/IAPP-binding mitochondrial protein, also called DIABLO, is a mitochondrially-associated protein which migrates to the cytosol upon the activation of apoptosis, facilitating this process by restricting the activity of apoptotic inhibitors[55]. Lowered levels of this protein in the plasma thus seem likely to reflect diminished levels of cell death in disease-relevant (or other) tissues. Coactosin-like protein, encoded by the *COTL1* gene and also called Coactosin-like F-actin binding protein, and CLP, is enriched in haematopoietic cells (BioGPS), and regulates leukotriene synthesis. Low levels may therefore reflect more limited inflammation in individuals whose disease deteriorates quickly. Testican-1, also called SPOCK1, SPARC, and Osteonectin is enriched in the brain. The function of SPOCK1, a $Ca^{2+}$ binding proteoglycan, is currently unknown, though roles in neuronal development[56] adipocyte differentiation[57] and as an extracellular matrix factor controlling epithelial to mesenchymal transition[58] have been suggested. Finally, HEMK2 (N6AMT1, PrmC) is a ubiquitously-expressed DNA methyl transferase[59].

To investigate causal associations, we undertook MR analysis of six of our identified protein biomarkers that were associated with diabetes progression. We repeated previous findings that GDF15 is causally associated with diabetes progression[32] and found a possible causal association for IL-18Ra and FAS. These data suggest that these two proteins are likely to be causally associated with diabetes progression. The lack of a causal association of NogoR with diabetes risk does suggest that NogoR may similarly not be causally associated with diabetes progression, however, our functional studies do suggest a causal mechanism linking NogoR with abnormal glucose metabolism.

This study has several limitations. Between cohorts there was heterogeneity, and this was in part due to the nature of the cohorts, for example the ACCELERATE cohort is a clinical trial which is different in setup than the discovery studies. Nonetheless, the results of the study were robust, for example a large number of biomarkers were also found to be associated with prevalent and incident diabetes. A second limitation is that for some protein biomarkers ELISA assays are currently unavailable to validate the signals obtained in the SomaLogic screen. Thirdly, the interactions between metabolite classes was not explored in detail, and future studies may generate groups of biomarkers which when considered together collectively provide improved predictions of disease progression[60]. Finally, in the Mendelian Randomization analysis we could only investigate the causal association with diabetes risk, and some of the genetic instruments were weak, so the absence of causality in MR analysis does not mean that our findings with diabetes progression were non-causal. Indeed,

plausible evidence for causality was subsequently obtained in functional studies in preclinical models for NogoR and IL-18Ra.

Our findings highlight molecular changes that associate with glycaemic deterioration once diabetes has developed. Importantly, we describe biomarkers for type 2 diabetes progression of different chemical classes. We provide direct functional analyses implicating two of these (NogoR, IL-18Ra) as likely to contribute directly to disease progression. By better understanding the biological drivers of glycaemic deterioration in diabetes it may be possible to target therapies to these processes to prevent or slow diabetes progression, potentially transforming diabetes treatments.

## Methods

### Ethical considerations

**Discovery cohorts.** This study the study was conducted in line with the Declaration of Helsinki[61]. For DCS, the Ethical Review Committee of the VU University Medical Center, Amsterdam approved the study and written informed consent was obtained. The Tayside Medical Ethics Committee approved the GoDARTS study and participants provided written informed consent. The ANDIS protocol was approved by the Regional Ethical Review Board in Lund, Sweden (584/2006, 2011/354, 2014/198). All participants provided written informed consent.

**Replication cohorts.** The MDC study was approved by the Regional Ethical Review Board in Lund, Sweden (LU 51/90). All participants provided written informed consent. The DESIR study was approved by the Ethics Committee (CCPPRB) of the Bicêtre Hospital and all participants provided written informed consent. Plasma sample and data used in this study deriving from the ACCELERATE trial were provided by Eli Lilly pharmaceutical company (data owners of the placebo arm of the trial; co-authors IP, KD, AE)[62]. The multicentre trial involved 543 centres in 36 countries and for each of them the appropriate national and/or institutional regulatory and ethics boards approved the protocol independently (protocol number of internal approval at Lily, I1V-MC-EIAN-9/19/19)[62]. All patients provided written informed consent. Use of existing de-identified samples from the ACCELERATE cohort for the present research is classified as non-human research, and thus IRB approval was not required. The AGES-Reykjavik study was approved by the NBC in Iceland (approval number VSN-00-063), and the National Institute on Aging Intramural Institutional Review Board, and the Data Protection Authority in Iceland. All participants provided written informed consent. Study participants did not receive compensation for any of the studies included.

**Functional studies.** Human pancreatic islets were purchased from Prodo Labs through the Integrated Islet Distribution Programme, and used in Lilly Laboratories Indianapolis, IN, in compliance with the Eli Lilly Bioethics Framework for Human Biomedical Research, with appropriate research consent from the organ procurement organizations. According to regulation 45 CFR 46.of the U.S. Department of Health and Human Services (https://www.hhs.gov) this research is classified as not-human subject research and as such does not require IRB approval.

Ethics approval for rodent studies was obtained from the UK Home Office, according to the Animals (Scientific Procedures) Act 1986, with local ethical committee (Imperial College AWERB) and under a personal project license (PPL) number PA03F7F07 to I.L., or the Animal Care Committee at the Institut de recherches cliniques de Montréal (J.E.). Animals were maintained in approved institutional animal facilities overseen by qualified veterinary teams, under specific pathogen free conditions. Day to day monitoring of animal wellbeing was performed by facility technicians and by the researchers directly involved in the studies.

None of the cell lines used feature in the list of known misidentified cell lines (https://iclac.org/databases/cross-contaminations).

**Discovery cohorts.** Specific details on DCS[63], GoDARTS[64] and ANDIS[65] have been described elsewhere[66]. These cohorts were selected based in part on satisfactory quality control for biomarkers stability in stored samples.

**DCS.** The Hoorn Diabetes Care System (DCS) cohort is a prospective cohort with currently over 14,000 individuals with routine care data[63]. The Ethical Review Committee of the VU University Medical Center, Amsterdam approved the study. In 2008–2014, additional blood sampling was done in 5500 participants, who provided written informed consent. These samples were used for this study. The turbidimetric inhibition immunoassay for haemolyzed whole EDTA blood (Cobas c501, Roche Diagnostics, Mannheim, Germany) was used to measure HbA1c. HDL (mmol/L) was measured enzymatically (Cobas c501, Roche Diagnostics). C-peptide was measured on a DiaSorin Liaison (DiaSorin, Saluggia, Italy).

**GoDARTS.** The Genetics of Diabetes Audit and Research Tayside Study (GoDARTS) is a cohort of ~8,000 individuals with T2D. The study was approved by the Tayside Medical Ethics Committee and all individuals provided written informed consent. Laboratory measurements were measured in a non-fasted state. C-peptide was measured on a DiaSorin Liaison (DiaSorin, Saluggia, Italy).

**ANDIS.** In the All New Diabetics in Scania (ANDIS) cohort, people with incident diabetes within Scania County, Sweden were recruited from January 2008 until November 2016 and all participants gave written informed consent. Regional ethics review committee in Lund approved the study. An electro-chemiluminescence immunoassay was used to measure C-peptide on a Cobas e411 (Roche Diagnostics, Mannheim, Germany) or a radioimmunoassay (Human C-peptide RIA; Linco, St Charles, MO, USA; or Peninsula Laboratories, Belmont, CA, USA). The Clinical Chemistry database was used to obtain HbA1c levels.

**Validation cohorts.** External validation was performed in four external cohorts, ACCELERATE, AGES-Reykjavik, MDC-CC and DESIR. ACCELERATE is a clinical trial aimed at investigating the effect of evacetrapib on major adverse cardiovascular outcomes and has been described elsewhere[62]. For the current study we only included the 6,054 individuals in the untreated arm. From this group, we selected 2,978 individuals with type 2 diabetes. In this group, 1,003 individuals were excluded that did not have C-peptide levels or HbA1c levels, 72 were excluded because the age at diagnosis was <35 years, 31 were excluded because they were on insulin at baseline and 22 were excluded because they had >2 non-insulin glucose-lowering drugs and HbA1c levels > 8.5%. The final set consisted of 1,850 individuals of which 162 reached the primary endpoint.

AGES-Reykjavik is a prospective population-based study from Iceland[32,67]. In fasted blood samples protein levels were measured with the Somalogic platform. At baseline there were 4784 individuals free of diabetes and 654 with type 2 diabetes. Of 2940 individuals free of diabetes at baseline and with 5-year follow-up information, 112 developed type 2 diabetes[32].

Malmö Diet and Cancer Cardiovascular Cohort (MDC-CC) is a population-based cohort comprised of people living Malmö[68]. Metabolites were measured in 3423 individuals of which 402 developed type 2 diabetes. Lipids were measured using the Lipotype platform in 3667 individuals of which 555 individuals developed type 2 diabetes[69]. Proteins were measured using the Olink Proseek Multiplex proximity extension assay in 4915 individuals of which 700 developed type 2 diabetes.

DESIR is a prospective population-based cohort comprised of middle-aged European individuals. Metabolomics was measured by Metabolon (Durham, NC)[70]. In DESIR there were 43 prevalent and 231 incident cases and 813 controls.

**Molecular measurements.** A flowchart of the current study is given in the Supplementary Fig. 1. In the three discovery cohorts, those were selected with an age at diagnosis >35 years, not GAD positive, with GWAS data and with a blood sample close to diagnosis, with a median diabetes duration of 2.6, 1.4 and 0 years in DCS, GoDARTS and ANDIS respectively (Table S1). The metabolomics, lipidomics and proteomics groups were of different sizes (see further details below). Individuals were ranked based on the time between diagnosis and sampling date and those with the smallest time between diagnosis and sampling were selected. For metabolomics, we selected 1267 in DCS, 900 in GoDARTS and 900 in ANDIS. For lipidomics, 900 individuals were selected in DCS, GoDARTS and ANDIS. For proteomics, we selected 600 individuals in DCS and GoDARTS.

**Small charged molecule analytes.** LC-MS grade water ($H_2O$), methanol (MeOH), isopropanol (IPA), and acetonitrile (ACN) were purchased from Honeywell International Inc. (Morristown, NJ, USA). HPLC grade dichloromethane (DCM), anhydrous ACN, analytical grade formic acid (HCOOH), and reagent grade potassium carbonate ($K_2CO_3$), potassium bicarbonate ($KHCO_3$), sodium hydroxide (NaOH), hydrochloric acid (HCl), and 5-sulfosalicylic acid dihydrate (SSA) were purchased from Sigma-Aldrich (Steinheim, Germany). 6-Aminoquinoline-N-hydroxysuccinimidyl carbamate for amino acid derivatization was obtained from Santa Cruz Biotechnology, Inc. (Dallas, TX, USA).

Small charged molecules (referred to as metabolomics) were analysed in plasma samples from 2,973 individuals, including aminoadipic acid, alanine, citrulline, glutamic acid, glutamine, glycine, glycocholic acid, glycoursodeoxycholic acid, homocitrulline, indoxyl sulfate, isoleucine, kynurenine, leucine, phenylalanine, symmetric dimethylarginine / asymmetric dimethylarginine, taurine, taurocholic acid, tryptophan and tyrosine. The samples were stored at −80 °C and extracted on ice. Sample were extracted using a modified Folch procedure. Nineteen heavy-labelled pure standards were spiked into each sample, deuterated molecules were purchased as previously described[71]. Samples were randomized t be included in sequences of 9, quality control (QC) samples consisted of two types, firstly a standard reference plasma (1950 metabolite human plasma from NIST) was added for every tenth sample followed by a blank sample. A second QC were pooled samples and calibration curve samples, consisting of a dilution series of pure reference standards for each of the 19 measured compounds, were added at the start and at the end of every 100 samples.

Absolute quantitation of the 19 small charged molecules was performed using UHLPC-MS/MS (UHLPLC: 1290 Infinity system from Agilent Technologies, Santa Clara, CA, USA; MS/MS: 6460 triple quadrupole system from Agilent Technologies) with a Kinetex® F5 column (100 × 2.1 mm, particle size 1.7 μm) from Phenomenex (Torrance, CA, USA) at flow a rate of 0.4 mL/min and an injection volume of 5 μL. The elution mobile phases consisted of (A) $H_2O$ + 0.1% HCOOH (A) and (B) ACN:IPA (2:1, v/v) + 0.1% HCOOH. The gradient started with 1% B for the first minute. With mobile phase B increased from 1−18% over 1−1.8 min, from 1.8 to 3.4 min 18−21% B, from 3.4 to 7 min 21−65% B, from 7 to 7.1 min 65−100% B and from 7.1 to 8.9 min 100% B. The capillary voltage was set to 3000 V and the nozzle voltage to 1000 V. MS- and MS/MS-spectra (scan range m/z 40−600) were acquired using selected reaction monitoring (SRM). Each analyte was previously optimized for the best precursor, product ion as well as optimal fragmentor voltage and collision energy[71].

Post-processing peak identification, normalization and quantification was carried out in MassHunter B.06.01 software by Agilent Technologies. Normalisation was performed using the heavy-labelled internal standards and quantification was carried out by matching to the pure standard calibration curves.

In DCS, all samples passed QC and were used in the analysis. In GoDARTS, three failed QC and the remaining samples were used for analysis. In ANDIS, 4 failed QC and of the 892 remaining samples, 811 were free of the outcome at sampling. In addition, a validation set was generated comprised of 2668 individuals (699 GoDARTS, 1,969 ANDIS).

**Lipid measurements.** Data of 614 lipids were generated using a QExactive mass spectrometer (Thermo Scientific) equipped with a TriVersa NanoMate ion source (Advion Biosciences) on the Lipotype lipidomics platform (Lipotype, Dresden, Germany)[72]. The Lipotype Shotgun lipidomics method is reported according to guidelines of the "Lipidomics Standardization Initiative" (https://doi.org/10.1038/s42255-022-00628-3) and can be found in Supplemental Material. Lipid nomenclature is used and SwissLipids database identifiers are provided (Supplementary Data 3)[73]. Samples ($n = 2,608$) were measured in batches of 84 samples each. Lipid identifications with a signal-to-noise ratio >5, and a signal intensity 5-fold higher than in corresponding blank samples were considered for further data analysis. Validation of identity between cohorts was achieved by confirming that each TAG species was identified with a similar fatty acid profile. Spectra were analysed with in-house developed lipid identification software based on LipidXplorer[74]. TAGs are quantified as species (e.g. TAG 48:1;0). Fatty acid amounts within TAG species were calculated based on intensities of neutral losses of fatty acid fragments. Only fatty acids that were measured in at least one cohort in 80% of the subjects were considered. Profiles were standardized on each species such that fatty acid amounts within one species sum up to 100% within every subject. From this data mean and standard deviations across each cohort were calculated (Supplementary Data 4). Eight reference samples were used to apply batch correction and amounts were further adjusted for analytical drift ($p$-value slope ≤ 0.05 and $R^2 ≥ 0.75$ and the relative drift > 5%). The reference samples are replicates from a pool of plasma purchased from the Deutsches Rotes Kreuz, Kreisverband Dresden e.V. After quality control 162 lipid species were used in this study. The median coefficient of subspecies variation of the 162 lipids used as accessed by reference samples was 9.49% across all three cohorts. In DCS, 900 individuals were included for lipidomics measurements, all passed QC, and all were suitable for analysis. In GoDARTS, 898 individuals were included in the analysis, 1 failed QC and all 897 remaining samples were included in the analysis. In ANDIS, 896 individuals were included in the analysis, 5 failed QC and of the 891 remaining samples, 811 were free of the outcome at sampling.

**Protein measurements.** Proteins were measured on the SomaScan® Platform from Somalogic ($n = 1195$ proteins) on the SomaLogic SOMAscan platform (Boulder, Colorado, USA) in 1188 individuals. Top associated proteins were validated in ANDIS ($n = 1992$) and ACCELERATE ($n = 1850$) using ELISA with time to insulin requirement as the outcome. External validation was performed for the top proteins based on $P$-value and/or effect size for which an ELISA was available or could be developed. Aptamers used for the protein measurements were considered specific if a protein QTL was presented in cis[75,76]. SNPs were considered when the minor allele frequency was ≥1% and 1 Mb from transcription start or end site. The primary set used was that of Ferkingstad et al.[76]. given the large population, but when aptamers were not included in that particular set, the Sun et al. study was used[75].

**Primary endpoint.** The primary endpoint time to insulin requirement was defined as the period from diagnosis to a clinical endpoint of the

earlier of (i) starting sustained (>6 months duration) insulin treatment or (ii) clinical requirement of insulin as indicated by two or more HbA1c measurements >8.5% > 3 months apart when on two or more non-insulin diabetes therapies[77]. In DCS, 600 individuals were included for proteomics measurements, 11 failed QC and all were included for analysis. In GoDARTS, 600 individuals were included in the analysis, 1 failed QC and the 599 remaining samples were included in the analysis.

**Federated database.** All main analyses were performed on a federated database system. Opal, an open-source data warehouse (Open Source Software for BioBanks, OBiBa) was used to store cohort data on local nodes and remote analysis was performed in R using *DataSHIELD*[78] and *dsSwissKnife* R packages[79]. A central server was set up at the Swiss Institute of Bioinformatics to manage federated node access, user administrator and software deployment. Local nodes were set up at the respective cohorts. All data was harmonized according to the CDISC Study Data Tabulation Model (www.cdisc.org) prior to inclusion into the federated database.

**Mendelian randomisation.** Genetic instruments for proteins, lipids and metabolites predictive of progression in the three cohorts were obtained from published GWASs. Protein quantitative trait loci (QTLs) were obtained from Gudmundsdottir et al.[32]. Lipid QTLs were obtained from Tabassum et al.[80]. Metabolite QTLs were obtained from Lotta et al.[81]. Only QTLs with $P$-values < $5 \cdot 10^{-8}$ were included. For traits with one instrument Wald ratio was used and for multiple instruments inverse variance weighting. Instruments were excluded when in LD ($r^2 > 0.1$). Genetic instruments for type 2 diabetes were obtained from the latest GWAS on incident type 2 diabetes[82]. Horizontal pleiotropy was estimated based on MR-Egger intercept. Cochran Q-statistic was used to estimate heterogeneity of instruments.

**Cells and cell culture.** HEK-Blue IL-18 cells passages 1-16 (InvivoGen, USA, #Cat code: hkb-hmil18) were cultured in 4.5 g/L glucose Dulbecco's Modified Eagle's Medium (DMEM) supplemented with 10% foetal bovine serum (FBS), 2 mM L-Glutamine, 50 U/ml penicillin, 50 µg/ml streptomycin (Sigma, UK) and 100 µg/ml Normocin (InvivoGen, USA). HEK-Blue IL-18 cells were designed to detect bioactive IL-18 by monitoring the activation of the NF-κB and AP-1 pathways. They were generated by stable transfection of HEK293 derived cells with the genes encoding IL-18R and IL-18 receptor accessory protein (IL-18RAP). Additionally, the TNF-α and the IL-1β responses have been blocked to guarantee a specific respond to IL-18. Cells were seeded in T75 flasks and sustained at 37 °C in a humidified incubator containing 5% CO2. Experiments were carried out with HEK293 cells as well as a negative control.

**Cell transfection.** HEK-Blue IL-18 and HEK293 cells were seeded in 12-well plates at 30,000 per well and transfected after 48 h with both NF-κB and Renilla using Lipofectamine 2000 DNA Transfection Reagent (Invitrogen, USA). For each well, 1.5 µl of Lipofectamine reagent was diluted in 100 µl Opti-MEM Gibco medium (Thermo Fisher, USA). The amount of 1 µl of NF-κB DNA at a concentration of 472.6 ng/µl and 1 µl of Renilla at a concentration of 40 ng/µl were also diluted in 100 µl of Opti-MEM medium for each well, the diluted DNA mix was then added to diluted Lipofectamine reagent and the transfection mix was incubated for 20 min at RT. Cells were washed with 1 ml of PBS and 400 µl of Opti-MEM medium was added per well. Cells were transfected with 200 µl transfection mix each well. Medium was then changed to complete growth medium (DMEM supplemented with 10% FBS, 2 mM L-Glutamine, 50 U/ml penicillin, 50 µg/ml streptomycin and 100 µg/ml Normocin) after 4 h.

**Cell treatment and stimulation.** Transfected cells were stimulated with different concentrations of IL-18, IL-18Rα or both to measure

changes in NF-κB activation. Concentrations were made by diluting different volumes of 1 µM IL-18 and IL-18Rα stock (diluted from freeze-dried powder in distilled water) in serum-free medium to exclude the effect of serum factors and cells were stimulated with 300 µl of each condition per well of 12-well plate for 6 h. Recombinant human IL-18 and rhIL-18 Rα/Fc chimera were obtained from MBL and R&D Systems respectively. Normal rabbit IgG from Abcam (ab171870) was used as a negative control.

**Dual luciferase reporter assay.** Cells transfected with NF-κB and stimulated with IL-18/IL-18Rα were lysed following 6 h of treatment by adding 200 µl of Promega Passive Lysis Buffer (PLB) to each well and gentle shaking for 15 min at RT. Cell lysates were stored at −20 °C and luciferase activity was measured the next day using Promega Dual-Luciferase Reporter Assay System according to manufacturer's instructions to study gene expression at the transcriptional level. Briefly, Luciferase assay reagent II (LAR II) was dispensed into a luminometer tube for each condition. Cell lysate was resuspended in tube containing LAR II and Firefly luciferase activity was measured using Berthold Lumat LB 9507 Tube Luminometer. Next, Stop & Glo reagent was added to each tube and the Renilla luciferase activities were measured. Measurements were read with PuTTY software and all Firefly luciferase activities were normalized with Renilla luciferase activities to obtain the NF-κB signalling ratios using Microsoft Excel software.

**Hepatocyte cell isolation and culture.** Ethics approval for rodent studies was obtained from the Animal Care Committee at the Institut de recherches cliniques de Montréal (JE). Male C57B/6 N mice (Taconic Biosciences, East Greenbush, N.Y.) were maintained on a 12-h light-dark cycle and fed regular grain-based diet (GBD) (Teklad Global 18% Protein Rodent Diet) from Teklad Diets, Envigo (Huntingdon, UK) consisting of 24 kcal% protein, 58 kcal% carbohydrates, and 18 kcal% fat. Animals were euthanized using isofluorane and CO2. For each experiment, primary hepatocytes from 2–3 mice (aged 10–12 weeks) were isolated using collagenase (Liberase) perfusion and Percoll gradient purification. Cells were plated at a density of 450,000/well in 6 well plates in DMEM medium containing 10% FBS, 2 mM sodium pyruvate, 1 µM dexamethasone, 1% Pen/Strep, and 0.1 µM insulin. 2 hrs later, medium was exchanged with DMEM maintenance medium supplemented with 0.2% BSA, 2 mM pyruvate, 1% Pen/Strep, 100 nM dexamethasone, and 1 nM insulin. Cells were incubated in similar medium lacking dexamethasone and insulin for 24 h prior to treatment insulin and/or NogoR treatment.

Murine C3H10T1/2 cells (ATTC, CCL-226™, courtesy Dr Pierre Moffat, McGill University, Montreal, QC, Canada) were seeded at 100,000 cells/well in 12 well dishes in DMEM/F12 media supplemented with 10% FBS, 1% P/S. After 48 h, when cells reached 100% confluency, media was replaced with differentiation media DMEM/F12 (10% FBS/ 1% P/S) containing 2 µg/mL Insulin, 0.5 mM IBMX, 2 ug/mL Dexamethasone, and 5 µM Rosiglitazone for 48 hrs. Media was then replaced with maintenance media containing DMEM/F12 (10% FBS/ 1% P/S) supplemented with 2 µg/mL insulin for 5 days. Cells were then starved overnight with media lacking FBS and insulin.

HepG2 Cells (human hepatocyte line, courtesy Pr. Axel Kahn, Institut Cochin, Paris, France) were cultivated in DMEM media with 1 g/L of glucose, supplemented with 10% SVF and maintained at 37 °C and 5% CO2. Cells were plated in DMEM media with 10% SVF.

**Cell treatment and western (immuno-) blotting.** Primary hepatocytes and C3H10T1/2-derived adipocytes were treated with NogoR recombinant protein (1, 10, 100 nM) for 3 or 6 h respectively prior to 100 nM insulin stimulation for 15 min. HepG2 cells were treated with 0, 1 nM, 10 nM, or 100 nM of NogoR or CRELD1 (Mouse Fc tagged, Sino Biological, Cat # 51149-M02H) recombinant proteins for 3 h and then

stimulated with insulin for 15 min. Proteins were then extracted (primary hepatocytes were lysed with RIPA cell lysis buffer (50 mM Tris HCl PH 8, 150 mM NaCl, 1% NP-40, 0.5% sodium deoxycholate, 0.1% SDS)) supplemented with phosphatase inhibitors (Thermofisher A32957) and protease inhibitors (Roche 11697498001). Adipocytes were lysed with RIPA buffer from Thermofisher Scientific (cat. ner 89900) (25 mM Tris HCl pH 7.6, 150 mM NaCl, 1% NP-40, 1% sodium deoxycholate, 0.1% SDS) supplemented with Halt™ Protease Inhibitor Cocktail (100X) from thermo scientific (reference 78430) and phosphatase inhibitors (10 mM NaF, 1 mM $Na_3VO_4$, 1 mM sodium pyruvate, 10 mM β-glycerophospate). HepG2 were lysed with RIPA buffer (20 mM Tris-HCl, 50 mM NaCl, 1 mM Na2-EDTA, 1 mM EGTA, 1% NP-40, 1% sodium deoxycholate) with 1% Phosphatase Inhibitor Cocktails (P0044 and P5725, Sigma-Aldrich) and 1% Protease Inhibitor Cocktail (P8340, Sigma-Aldrich). Protein concentration was measured and proteins were loaded and separated on SDS-PAGE then transferred to nitrocellulose or PVDF membranes. Membranes were then blocked in 3–4% BSA TBS-T 0.1% or 5% milk TBS-T 0.1% for 1 h and further incubated with the proper primary antibody in 3% BSA TBS-T 0.1% overnight at 4 °C (pAkt (Ser473) from cell signalling (9271 S or 4060 S) or pAkt (Thr308) from cell signalling (9275 S), Akt from cell signalling (9272 S), Insulin receptor β from cell signalling (3025 S), Phospho-IGF-I Receptor β (Tyr1131)/Insulin Receptor β (Tyr1146) from cell signalling (3021 S), and Beta actin from Thermofisher A5441) or from Cell Signalling (3700 S) and was used a t dilution of 1:5000. The membranes were washed 3 times with TBS-T 0.1% and incubated with the proper secondary antibody diluted at 1:2500 for 1 h at RT (for primary hepatocytes; anti-rabbit Li-cor, C81106-05 or anti-mouse Li-cor, D10603-01 were prepared in 3% BSA TBS-T 0.1%) while horseradish peroxidase (HRP)-conjugated secondary antibodies (prepared in 3–4% BSA TBS-T 0.1%) were used for adipocytes and HepG2 cells at a dilution of 1:2000 for the Rabbit antibody and 1/5000 for the mouse antibody. Membranes were then washed 3 times in TBS-T 0.1% and developed.

**In vivo metabolic tests.** Adult male C57BL/6 J mice (Envigo, Huntingdon, UK) were maintained under controlled temperature (21–23 °C), humidity (45–50%) and light (12:12 h light–dark schedule, lights on at 0700 h) in specific pathogen-free (SPF) cages. Animals were screened for Federation of European Laboratory Animal Science Associations (FELASA) list pathogens. For oral glucose tolerance tests (OGTT), C56BL6J mice (Charles River, $n = 5$) maintained for 4 weeks on a high fat high sucrose (HFHS) diet (a 58 kcal% Fat and Sucrose diet (D12331, Research Diet, New Brunswick, NJ)) were fasted for 16 h prior to experiment and received an oral glucose load (2 g/kg of body weight). For insulin tolerance tests, mice received an intraperitoneal injection of insulin (1 IU/kg) after 3 h fasting ($n = 5$). Blood glucose levels were determined by tail venepuncture using a glucose meter (Accu-Chek; Roche, Burgess Hill, UK) at 0, 15, 30, 60 and 120 min. after the glucose load. For insulin measurements blood was collected in EDTA covered tubes at times 0, 15 and 30 min. after the glucose load (2 g/kg of body weight). Subsequently, blood was centrifuged at 4000 x g for 20 min. at 4 °C and plasma was collected. Insulin was determined by ELISA (CrystalChem, 90080), according to the manufacturer's instructions. Daily intraperitoneal injections of NogoR were performed with indicated dose of NogoR (mouse, His and Fc tag; Sino Biologicals Cat # 50106-M03H). Animals were euthanized by cervical dislocation.

**In vivo metabolic tests (db/db cohort).** A cohort of B6.BKS(D)-Leprdb/J mice (also known as db/db) was purchased from Jackson Laboratory (Stock No: 000697). At 8 weeks of age, all animals received a subcutaneous implantation of Alzet osmotic pump (model 1002) containing either NogoR (mouse, His and Fc tag; (Sino Biologicals Cat # 50106-M03H) at infusion rate 100 ng/day) or saline. At 4 weeks of continuous infusion, mice underwent an oral glucose tolerance test

(OGTT) by receiving an oral glucose load (2 g/kg, $n = 4$–5) following 16 h fasting. For insulin tolerance tests, mice received an intraperitoneal injection of insulin (1 IU/kg, $n = 4$–5) after 3 h fasting. Blood glucose levels were determined by tail venipuncture using a glucose meter (Accu-Chek; Roche, Burgess Hill, UK) at 0, 15, 30, 60, 90 and 120 min. after the glucose load. For insulin measurements, blood was collected in EDTA covered tubes at times 0, 15 and 30 min. after the glucose load (2 g/kg of body weight). Subsequently, blood was centrifuged at 4000 x g for 20 min. at 4 °C and plasma was collected. Insulin plasma concentration was determined by ELISA (CrystalChem, 90080), according to the manufacturer's instructions. NogoR plasma concentration was determined by ELISA (RayBiotech, ELM-NOGOR-1), according to the manufacturer's instructions.

**Immunostaining of pancreatic sections.** To measure β cell mass, whole pancreata were removed from 4–5 mice, placed in cold PBS and carefully all the surrounding fat and non-pancreatic tissue (e.g. intestine, lymph nodes, etc) were removed[83,84]. After removing excess buffer, pancreata were weighed and fixed in freshly prepared 10% formalin at room temperature for 24 h, followed by embedding in paraffin blocks. Longitudinal cross pancreatic sections were cut using a microtome at 5 μm thickness and collected at 30 μm intervals. At least four slides from each pancreas were processed for islet and beta-cell mass measurements. For beta-cell mass measurement, immunohistochemistry was done with anti-guinea pig insulin antibody (Agilent-DAKO, Santa Clara, CA) to mark β-cells and alkaline phosphatase conjugated second antibody (Jackson Immunoresearch) and finally developed with the Vector Red alkaline phosphatase substrate kit (Vector Laboratories). Harris-modified hematoxylin was used for counter-staining before mounting the slides with Vectamount medium (Vector Laboratories). The slides were scanned at 20X using a high resolution Aperio ScanScope model CS slide scanner (Leica Biosystems Inc., Concord, ON, Canada) to assess islet/β-cell area and the whole pancreas area via the Aperio Pixel count algorithm v9 (ImageScope v12.3.2.5030, Leica Biosystems Inc.), followed by calculation of the ratio of β-cell area to whole pancreas area. Beta-cell mass was calculated by multiplying the ratio of β-cell area to whole pancreas area with whole pancreatic masses (mg) measured before the fixation step. Morphometric measurements were performed by identifying manually regions of interest (ROIs) around insulin-positive islets. The surface of all islets from at least 4 slices (>400 islets) were calculated for each ROI (ImageScope) and used to generate the size frequency distribution (surface) profile.

**Insulin secretion from mouse and human islets.** Mouse pancreatic islets were isolated from male C57BL/6 mice (Envigo, Indianapolis, IN) by collagenase digestion. Use of animals was approved by Eli Lilly and Company's Institutional Animal Care and Use Committee. Human pancreatic islets from listed cadaver organ donors that were refused for pancreas or islet transplantation were obtained from Prodo Labs (Irvine, CA) and InSphero AG (Schlieren, Switzerland) and were used in accordance with internal review board ethical guidelines for use of human tissue. Next of kin consent was obtained where relevant. Islets were cultured in the complete PIM(S) Prodo Islet Media (Prodo Labs) and RPMI 1640 medium (Invitrogen) supplemented with 11 mm glucose, 10% (v/v) heat-inactivated foetal bovine serum (Invitrogen), 100 IU/ml penicillin, and 100 μg/ml streptomycin (Invitrogen).

For insulin secretion in mouse islets, islets were incubated for 30 min. in Earle's balanced salt solution (EBSS) buffer supplemented with 3 mM glucose and 0.1% BSA. Then groups of three islets were selected and cultured with tested proteins at indicated glucose concentration in 300 μL of EBSS for 60 min at 37 °C. At the end of incubation, the supernatant was collected and subjected to insulin analysis. To measure insulin secretion in human pancreatic islets, single islets placed in a GravityTRAP 96-well plate (InSphero) were washed and

incubated for 30 min in 100 μL of EBSS supplemented with 0.1% BSA and 3 mM glucose. Then the buffer was replaced with 100 μL of EBSS containing indicated glucose and protein concentrations and further cultured for 60 min at 37 °C. At the end of incubation, the supernatant was collected and submitted for insulin analysis. Insulin levels were determined with the Meso Scale Discovery (Gaithersburg, MD) electrochemiluminescence insulin assay.

For chronic incubation experiments, after overnight recovery human or mouse islets were cultured in 12 well plate (20–30 islets per well) in the RPMI-1640 culture media containing tested proteins for 72 h. At the end of incubation, islets were transferred into EBSS supplemented with 3 mM glucose and 0.1% BSA. Then, 1 h insulin secretion in response to elevated glucose in islets pre-treated with proteins was measured.

**Quantification of β-cell proliferation.** After overnight recovery, mouse or human islets were cultured for 72 h in 12-well plates (200-300 islets per well) in RPMI-1640 medium containing 5 mM glucose, 2% FBS, 10 μM EdU and tested proteins. Leucettine L41 (10 μM) was used as a positive control. At the end of incubation, islets were washed, dispersed into single cells with the Accutase solution (Sigma) and placed in 96 well plate coated the Cell-Tak Cell and Tissue Adhesive (Corning). Cells were fixed, permeabilized and stained with Click-iT EdU HCS assay (ThermoFisher), PDX1 antibody (ab47308, Abcam) and Hoechst 33342 (ThermoFisher) nucleic acid dye. Cell images were captured and analysed using InSight Imaging System (ThermoFisher).

**Islet cell apoptosis.** After 3–4 day culture, mouse or human islets were plated (1 islet per well) in the GravityTRAP 96-well plate (InSphero) in 100 μl/well RPMI-1640 medium containing 11 mM glucose and 1% FBS. To induce cell death, islets were treated with glucose (25 mM) and palmitic acid (300 μM palmitate conjugated with BSA), cytokine mixture (120 ng/ml TNF-α, 60 ng/ml IL-1B and 240 ng/ml IFN-γ, R&D Systems) or tunicamycin (0.03 μg/ml, Tocris). After 72-h incubation, islets were caspase activity was measured with the Caspase-Glo 3/7 Assay (Promega) according to manufacturer's protocol.

**TUNEL staining of human pancreatic islets.** Human pancreatic islets from three independent donors were cultured for 72 h with the recombinant cytokine mix or recombinant NogoR protein at concentrations used for the caspase assay. Islets were pelleted into Histogel (Thermo Scientific), processed and embedded into paraffin. Paraffin blocks were cut into sequential sections 4 μm apart. To maximize the number of islets stained, 3 non-sequential slides chosen from each block for staining and analysis. Slides were stained with antibodies for insulin (A0564, Agilent Dako, Santa Clara, CA), glucagon (PU039-UP, Biogenex, Fremont,CA) and with DAPI (Thermo Scientific). Cell death was determined with the TUNEL assay using ApopTag In Situ Apoptosis Detection Kit (EMD Millipore). Slides were scanned with the PANNORAMIC 1000 scanner (3DHistech, Budapest, Hungary) and images were processed with Visiopharm software (Westminster, CO) to determine % of TUNEL insulin positive nuclei.

**Statistical analyses.** A Cox proportional hazard model was used to identify molecular risk factors for time to insulin requirement in R (v3.6.0) remotely on each cohort federated node using the *dssCoxph* function in *dsSwissKnife*[79]. Data was log transformed and scaled before analysis. Missing data in the omics were not imputed, but instead individuals were excluded that specific biomarker. The proportionality assumption was assessed using the cox.zph function from the survival R package. In each of the three cohorts DCS, GoDARTS and ANDIS, three Cox proportional hazard models were performed:

Model 1: Biomarker, age, sex, BMI, biomarker
Model 2: Biomarker, age, sex, BMI, HDL, C-peptide, biomarker

Model 3: Biomarker, age, sex, BMI, HDL, C-peptide, diabetes duration, glucose-lowering drugs, biomarker

All three models were stratified for HbA1c (strata: <53 mmol/mol, 53-75 mmol/mol and >75 mmol/mol). The effect of diabetes duration was also investigated on itself, but this did not influence the results, with high correlation between effect sizes before and after adjustment: metabolites 0.96 (95% CI:0.89-0.98), lipids 0.95 (95% CI:0.93-0.96) and proteins 0.91 (95% CI:0.91-0.93). Results from the three cohorts were meta-analyzed using the metagen function from the meta R-package. Heterogeneity of across cohorts was assessed using the $I^2$ metric. P-values were adjusted for multiple testing using the Benjamini Hochberg procedure. An FDR P-value below 0.05 was considered significant. Reported confidence intervals were adjusted for multiple simultaneous confidence intervals. For this, instead of showing the 95% confidence intervals, the interval was based on the number of tests and significant hits at FDR < 0.05. Specifically, the adjusted confidence interval to be used was calculated as 1 − q*R/m, where q is the level at which the FDR is controlled (0.05), R the number of significant tests at 5% FDR and m the total number of tests performed[85].

For the validation in MDC-CC, lipids and proteins identified in the discovery cohorts were tested against incident diabetes using Cox proportional hazard model on a local machine adjusted for age, sex and BMI. In the AGES- Reykjavik cohort, identified proteins were tested against incident and prevalent type 2 diabetes using logistic regression, adjusted for age and sex. In DESIR, logistic regression adjusted for age, sex and BMI was used to test for an association between metabolite levels and prevalent and incident diabetes.

The number of acyl chain length and number of double bonds are important for the direction of effect for TAGs[9]. The number of double bonds and the acyl chain length was compared to the hazard ratio observed for TAGs associated with time to insulin initiation.

For the insulin secretion assay and the caspase assay the effect of protein exposure was compared to vehicle using one-way ANOVA. In the animal studies, Student's t-test was used to compare the NogoR group with the control group for body weight, OGTT, insulin tolerance test (ITT), plasma NogoR levels. Differences in pAKT, AKT, IR, pIR between groups were tested using ANOVA and Student's t-test.

Figures and meta-analysis were performed locally with R (v4.0.3). Figures were made using ggplot2 (v3.3.2). Analysis of cellular and metabolic data were performed using GraphPad Prism versions 7.0–9.0 (San Diego, CA, U.S.A.). Where p-values were below 0.001 these were re-estimated using the function *DunnettTest* from the R package *DescTools*.

### Reporting summary
Further information on research design is available in the Nature Portfolio Reporting Summary linked to this article.

## Data availability
*Discovery cohorts* Summary statistics of lipidomic, proteomic and metabolomic data is available from a Shiny dashboard available from: https://rhapdata-app.vital-it.ch. The generated metabolomic, lipidomic and proteomic data in DCS, GoDARTS and ANDIS are considered sensitive patient data and can therefore not be publicly available in compliance with the European privacy regulations governed by GDPR and according to limitations included in the informed consents signed by the study participants. Please see below information on how to request the data.

Metabolomics and lipidomics (DCS, GoDARTS, ANDIS) data are available upon request by contacting the senior authors (dr. LM 't Hart (lmthart@lumc.nl)), prof. dr. E.R. Pearson (E.Z.Pearson@dundee.ac.uk), prof. dr. ir. JWJB Beulens (j.beulens@amsterdamumc.nl) and. dr. G. Rutter (g.rutter@imperial.ac.uk). Requests should include name and contact details of the person requesting the data, which molecular data and clinical variables are requested and the purpose of requesting

the data. Requests will be subject to consideration by the steering committees of the three cohorts (DCS, ANDIS, GoDARTS) and the management board of RHAPSODY. Time frame for a response will be within 4 months. Data requests under agreement will be considered for purposes of reproducing the data and subject to appropriate confidentiality obligations and restrictions.

DCS and GoDARTS proteomics data: restricted access for the proteomics data can be obtained via the European Genome/Phenome archive under accession number EGAD00010002447. Requests via EGA will be forwarded to the corresponding authors and subjected to the same procedure and time frame as the metabolomic and lipidomic data as outlined above.

Replication data: Proteomics data of individuals with incident diabetes described in Gudmundsdottir et al.[32] were used for lookups of our protein top hits. Lipid top hits were compared to the lipid data of people with and without diabetes from Fernandez et al. The GWAS on lipids described by Tabassum et al.[80] was used to identify lipid QTLs (https://mqtl.fimm.fi). The GWAS data of Lotta et al.[81] was used to identify metabolite QTLs. The GWAS on type 2 diabetes from Mahajan et al.[82] was used to find diabetes risk variants.

Functional studies: Source data for functional studies are provided with this paper. Source data are provided with this paper.

## Code availability
R code used is available via GitHub: https://github.com/roderickslieker/RHAPSODY (https://doi.org/10.5281/zenodo.7529655).

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

## Acknowledgements

We acknowledge the support of the Health Informatics Centre, University of Dundee for managing and supplying the anonymised data. The authors wish to thank the Core-IT group of SIB Swiss Institute of Bioinformatics and in particular Jorge Molina for expert technical help in setting up and maintaining the federated database. G.A.R. thanks Claudio Elgueta Karstegl for technical support. The authors thank participants of the included cohorts. We are grateful to Mélanie Guévremont and the CRCHUM Cellular Physiology core facility for beta cell mass analyses. The ANDIS study was financed by Swedish governmental funding of clinical research (ALF), the Faculty of Medicine at Lund University, the Swedish Research Council project grant no. 2020-02191 and strategic research area grant no. 2009-1039 (EXODIAB), from the Swedish Foundation for Strategic Research IRC15-0067 (LUDC-IRC) and Vinnova Swelife. E.R.P. holds a Wellcome Trust New Investigator Award (102820/Z/13/Z). G.A.R. was supported by a Wellcome Trust Investigator Award (212625/Z/18/Z), MRC Programme grants (MR/R022259/1, MR/J0003042/1, MR/L020149/1), an Experimental Challenge Grant (DIVA, MR/L02036X/1), a Diabetes UK Project grant (BDA16/0005485) and Innovation Canada for a John R Evans Leader Award. G.E.L. was supported by a CIHR Project Grant (PJT-153144) and holds the Canada Research Chair in Adipocyte Development. This project has received funding from the Innovative Medicines Initiative 2 Joint Undertaking, under grant agreement no. 115881 (RHAPSODY). This Joint Undertaking receives support from the European Union's Horizon 2020 research and innovation programme and EFPIA. This work is supported by the Swiss State Secretariat for Education, Research and Innovation (SERI), under contract no. 16.0097. Va.G. is supported by the Icelandic Research Fund (grant no. 184845-051). The Hoorn DCS cohort was supported by grants from the Netherlands Organisation for Health Research and Development (113102006, 459001015). J.E. is supported by the CIHR (PJT-168853).

## Author contributions

R.C.S., L.A.D, J.W.J.B., L.M.t.H., E.R.P., and G.A.R. designed the study. G.A.R. and R.C.S. wrote the manuscript. R.C.S., L.A.D., H.F., G.A.B., M.A. performed the analyses. G.A.R. oversaw biomarker shipments and assays, and coordinated all functional work in preclinical models. M.S., M.B., L.M.t.H., A.A.W.A.v.H., P.J.M.E., J.W.J.B., E.A., L.G., L.A.D., E.R.P. provided patient samples for analysis. E.A., L.L.-N., H.M.-M., M.S., E.G., I.L., R.M., A.E., A.G., F.A.A., G.E.L., J.E., S.K.S., J.L.S., O.C. performed all studies on recombinant NogoR and CRELD1, E.A., M.S. work on IL-18Ra. I.D., D.K., F.B., D.M., A.N., M.I. set up a federated node system for data analysis. R.C.S., L.A.D., H.F., D.M.A., E.A., A.A., M.J.G., M.K., F.M., T.S., A.W., C.L.Q., M.I. were involved in the (clinical) data pre-processing and quality control. G.N.G., A.F., M.K.H., D.M.A., I.P., T.J.P., B.T., V.L., L.G., P.W.F., K.B., S.B., M.B., K.D., G.A.R. contributed to the data acquisition and project logistics. M.J.G., C.K., K.S. generated the Lipotype data. C.L.Q., A.A., P.R., A.W., T.S., F.M. generated the metabolomics data and/or were involved in the quality control. F.O., C.F., and O.M. acquired the MDC-CC data and performed the lipid and protein validation in the MDC-CC. M.C. and P.F. performed the metabolite validation in DESIR. Va.G., Vi.G. and L.L.J. performed the protein validation in the AGES-Reykjavik cohort. P.J.M.E. and AAWAvH acquired the data from the Hoorn DCS cohort. All authors contributed to the data interpretation. All authors critically revised the manuscript and approved the final version. R.C.S. and L.A.D. are the guarantors of the work.

## Competing interests

K.S. is the CEO of Lipotype GmbH. K.S. and C.K. are shareholders of Lipotype GmbH. M.J.G. is employee of Lipotype GmbH. GAR has received grant funding and consultancy fees from Sun Pharmaceuticals and Les Laboratoires Servier. M.K.H. is an employee of Janssen Research & Development, LLC. A.F. and I.P. are employees of Eli Lilly Regional Operations GmbH. The AGES-Reykjavik proteomics study was supported by the Novartis Institute for Biomedical Research, and protein measurements for the AGES-Reykjavik cohort were performed at SomaLogic. L.L.J. is an employee and stockholder of Novartis. PR (Peter Rossing) has received honoraria to Steno Diabetes Center Copenhagen for consultancy and teaching from Astellas, Astra Zeneca, Boehringer Ingelheim, Bayer, Novo Nordisk, Sanofi, Gilead and Vifor and research grants from Novo Nordisk and Astra Zeneca. The remaining authors declare no competing interests.

## Additional information

Roderick C. Slieker [1,2,36], Louise A. Donnelly[3,36], Elina Akalestou[4,36], Livia Lopez-Noriega[4], Rana Melhem[5], Ayşim Güneş[6], Frederic Abou Azar[5], Alexander Efanov[7], Eleni Georgiadou [4], Hermine Muniangi-Muhitu[4], Mahsa Sheikh[4], Giuseppe N. Giordano [8], Mikael Åkerlund[8], Emma Ahlqvist[8], Ashfaq Ali [9], Karina Banasik[10], Søren Brunak [10], Marko Barovic[11], Gerard A. Bouland [2], Frédéric Burdet[12], Mickaël Canouil [13], Iulian Dragan[12], Petra J. M. Elders[14], Celine Fernandez [8], Andreas Festa[15,16], Hugo Fitipaldi[8], Phillippe Froguel [13,17], Valborg Gudmundsdottir [18,19], Vilmundur Gudnason [18,19], Mathias J. Gerl [20], Amber A. van der Heijden[14], Lori L. Jennings [21], Michael K. Hansen [22], Min Kim[9,23], Isabelle Leclerc[4,5], Christian Klose[20], Dmitry Kuznetsov[12], Dina Mansour Aly[8], Florence Mehl [12], Diana Marek[12], Olle Melander[8], Anne Niknejad[12], Filip Ottosson [8,24], Imre Pavo[15], Kevin Duffin[7], Samreen K. Syed[7], Janice L. Shaw[7], Over Cabrera[7], Timothy J. Pullen [4,25], Kai Simons[20], Michele Solimena[11,26], Tommi Suvitaival[9], Asger Wretlind [9], Peter Rossing [9,27], Valeriya Lyssenko[28,29], Cristina Legido Quigley[9,23], Leif Groop [8,30], Bernard Thorens [31], Paul W. Franks [8,32], Gareth E. Lim[5], Jennifer Estall [6], Mark Ibberson [12], Joline W. J. Beulens [1,33], Leen M 't Hart [1,2,34,37] ✉, Ewan R. Pearson [3,37] ✉ & Guy A. Rutter [4,5,35,37] ✉

[1]Department of Epidemiology and Data Science, Amsterdam Public Health Institute, Amsterdam Cardiovascular Sciences, Amsterdam UMC, location VUMC, Amsterdam, the Netherlands. [2]Department of Cell and Chemical Biology, Leiden University Medical Center, Leiden, the Netherlands. [3]Population Health & Genomics, School of Medicine, University of Dundee, Dundee, UK. [4]Section of Cell Biology and Functional Genomics, Division of Diabetes, Endocrinology and Metabolism, Department of Metabolism, Digestion and Reproduction, Imperial College London, London, UK. [5]CHUM Research Centre and University of Montreal, Montreal, QC, Canada. [6]IRCM and University of Montreal, Montreal, QC, Canada. [7]Lilly Research Laboratories, Eli Lilly and Company, Indianapolis, US. [8]Department of Clinical Sciences, Lund University, Malmö, Sweden. [9]Steno Diabetes Center Copenhagen, Gentofte, Denmark. [10]Novo Nordisk Foundation Center for Protein Research, Copenhagen, Denmark. [11]Paul Langerhans Institute Dresden (PLID) of the Helmholtz Center Munich at the University Hospital Carl Gustav Carus and Medical Faculty, Dresden, Germany. [12]Vital-IT Group, SIB Swiss Institute of Bioinformatics, Lausanne, Switzerland. [13]INSERM U1283, CNRS UMR 8199, European Genomic Institute for Diabetes (EGID), Institut Pasteur de Lille, University of Lille, Lille University Hospital, Lille F-59000, France. [14]Department of General Practice and Elderly Care Medicine, Amsterdam Public Health Research Institute, Amsterdam UMC–location VUmc, Amsterdam, the Netherlands. [15]Eli Lilly Regional Operations GmbH, Vienna, Austria. [16]1st Medical Department, LK Stockerau, Niederösterreich, Austria. [17]Division of Systems Biology, Department of Diabetes, Endocrinology and Metabolism, Imperial College London, London, UK. [18]Faculty of Medicine, University of Iceland, Reykjavik, Iceland. [19]Icelandic Heart Association, Kopavogur, Iceland. [20]Lipotype GmbH, Dresden, Germany. [21]Novartis Institutes for Biomedical Research, Cambridge, MA 02139, USA. [22]Cardiovascular and Metabolic Disease Research, Janssen Research & Development, Spring House, PA, USA. [23]Institute of Pharmaceutical Science, Faculty of Life Sciences and Medicines, King's College London, London, UK. [24]Section for Clinical Mass Spectrometry, Danish Center for Neonatal Screening, Department of Congenital Disorders, Statens Serum Institut, Copenhagen, Denmark. [25]Department of Diabetes, Guy's Campus King's College London, London, UK. [26]Molecular Diabetology, University Hospital and Medical Faculty Carl Gustav Carus, TU Dresden, Dresden, Germany. [27]Department of Clinical Medicine, University of Copenhagen, Copenhagen, Denmark. [28]Department of Clinical Science, Center for Diabetes Research, University of Bergen, Bergen, Norway. [29]Genomics, Diabetes and Endocrinology Unit, Department of Clinical Sciences Malmö, Lund University Diabetes Centre, Skåne University Hospital, Malmö, Sweden. [30]Finnish Institute of Molecular Medicine, Helsinki University, Helsinki, Finland. [31]Center for Integrative Genomics, University of Lausanne, CH-1015 Lausanne, Switzerland. [32]Department of Nutrition, Harvard School of Public Health, Boston, MA, USA. [33]Julius Center for Health Sciences and Primary Care, University Medical Center Utrecht, Utrecht, the Netherlands. [34]Department of Biomedical Data Sciences, Section Molecular Epidemiology, Leiden University Medical Center, Leiden, the Netherlands. [35]Lee Kong Chian School of Medicine, Nanyang Technological University, Singapore, Singapore. [36]These authors contributed equally: Roderick C. Slieker, Louise A. Donnelly, Elina Akalestou. [37]These authors jointly supervised this work: Leen M. 't Hart, Ewan R. Pearson, Guy A. Rutter. ✉e-mail: lmthart@lumc.nl; E.Z.Pearson@dundee.ac.uk; g.rutter@imperial.ac.uk

