## [Peer review file · Nature Communications]

Reviewers' comments:

Reviewer #1 (Remarks to the Author):

Novel biomarkers for glycaemic deterioration in type 2 diabetes: an IMI RHAPSODY study

Slieker et. al., leverage several large European population cohorts to identify metabolomic, lipidomic, and proteomic biomarkers of T2D disease progression (defined by sustained insulin therapy or deterioration of HbA1c despite ≥ 2 non-insulin diabetes therapies). Five metabolites were nominally associated with the primary outcome after adjustments for age, sex, and BMI ($n=2,975$; n cases =612) including homocitrulline, aminoadipic acid, isoleucine, glycocholic acid, and taurocholic acid. Only aminoadipic acid and homocitrulline were significant after adjusting for multiple hypothesis testing. These associations were directionally consistent but not significant in the replication cohorts ($n = 2,668$; n cases =668). For lipids, 8 triglyceride species were associated with progression and SM 42:2;2 was protective ($n = 2,608$; n cases = 486). These findings were not validated, but all were positively associated with incident diabetes in MDS. Ninety-eight proteins were nominally associated with the primary outcome ($n = 1,188$; n cases = 271), of which MIC-1/GDF-15 replicated in 1 of the validation cohorts and NogoR replicated in the other replication cohort. Mendelian randomization supported a possible causal association of PE 18:0,0-18:2;0 and the aptamers GDF-15, IL-18Ra, and FAS with T2D. Functional studies for a select group of proteins (MIC-1/GDF-15, IL-18Ra, NogoR, CRELD1, FAS, and ENPP7) suggest these proteins do not affect insulin secretion in mice or human islet cells. IL-18Ra and NogoR increases apoptosis of mouse islet cells, but at super-physiologic doses. NogoR infusions over 2 weeks improved glucose tolerance in mice on HFHS chow and HepG2 cells incubated with NogoR had increased insulin-induced signaling phosphorylation products. IL-18Ra was also found to attenuate IL-18 signaling in a dual luciferase reporter cell assay. In summary, a number of metabolomic, lipidomic, and protein biomarkers were identified associated with diabetes progression, some of which may be causally associated to T2D development via MR analysis and preliminary functional studies suggest that NogoR may affect insulin signaling and IL-18Ra could exert its effects by suppressing IL-18 signaling.

A strength of this manuscript is the leveraging of omics technology to identify biomarkers of T2D progression—which is an important clinical question that has not been previously studied in such large numbers of individuals. They also followed up their biomarker findings with preliminary functional studies to begin to elucidate biological mechanisms. Limitations are that only a few of these biomarkers actually replicate for the primary endpoint despite relatively large numbers and cases in the validation cohorts which diminishes the impact of these findings. Secondly, the primary outcome that was studied was diabetes progression while the functional studies were conducted in euglycemic mice (albeit fed a HFHS diet) and cells that had normal glucose homeostasis and insulin signaling. Thus, conclusions are more related to how these proteins potentially contribute to T2D development rather than their contribution to disease progression which is arguably the main focus of the paper.

Major critiques:

1. In the Molecular Measurements section, it states sample collection occurred between 0-3 years of diagnosis. What is the mean/median years between time of diagnosis to the lab draw? Were any sensitivity analyses conducted, especially if there was significant variability in this time, to determine how this could influence the primary analyses?
2. Fasting plasma glucose and HbA1c are known predictors of T2D progression, how strongly are the identified biomarkers of progression correlated to these clinical measures? Are there effects attenuated after adjustments for these values?
3. It is increasingly acknowledged that there are subtypes of T2D that are associated with different rates of progression to insulin therapy and complications. Have you considered looking at if BMI and c-peptide levels are effect modifiers of these associations with T2D progression?
4. Incident and prevalent diabetes associations were reported for some but not all the metabolites discussed in the results section, what were the associations for the other metabolites or were they not measured in MDC and DESIR?
5. In lipidomics a frequent challenge is successful separation of related lipid species (i.e., different TAG species often cluster together making it difficult to differentiate between those with similar acyl chain lengths). Given the top TAG findings all have similar acyl carbon lengths, how confident are you of the precision of these lipid measurements? Also given there were 3 cohorts with available lipidomics data, could one of them have been used for validation of the primary findings?
6. For the proteomics validation please show the meta-analysis results for the ACCELERATE and ANDIS trial in Figure 3. Also, were HSP 908, SMAC, Coactosin-like protein, Testican-1, and HEMK2 not measured in either validation cohorts?
7. As alluded to in the limitations, specificity of aptamer measurements in the SomaLogic screen is always a concern. However, sincere genetic data is available, were the reported aptamers associated with disease progression supported by GWAS hits that mapped to pQTLs?
8. Given the focus of the paper on biomarkers of diabetes progression, were any biomarkers associated with progression that were not associated with incident diabetes?
9. NogoR was shown to cause beta cell apoptosis at likely supra-physiologic doses but in healthy beta cells. It would be interesting to see NogoR exposure in beta cell models of diabetes.
10. Arguably, long term exposure to elevated levels of metabolites and proteins associated with incident disease and disease progression are what cause these outcomes. Have efforts been made to study animal models of more chronic exposures to elevated levels of NogoR?
11. Based on the functional study results, IL18-Ra should be protective for diabetes and disease progression given it blocks the inflammatory effects of IL-18. However, this is opposite of positive association found in the population cohorts. What is the proposed reason for this discrepancy in findings?
12. In the conclusions section, given the functional studies were all conducted in animal and cell models with normal glycemia it is premature to conclude that these findings provide “new mechanistic insights”

into glycemic deterioration after the development of diabetes. This section should be revised to reflect more accurately what was actually studied.

Minor critiques:

1. There are discrepancies in aptamer nomenclature throughout the manuscript that should be corrected (ex. GDF15 which is also MIC-1 is referred as GDF15 in the text, GDF15/MIC-1 in Fig 3, and MIC-1 in Supplemental T6).
2. In Figure 4, what is the difference between 4a and 4b?
3. In the limitations section (page 28, line 676) "A third limitation..." should actually be a second limitation

Reviewer #2 (Remarks to the Author):

Comments:

Remarks to the Author

In this manuscript, Roderick C Sliker et al use many diabetes individuals to investigate novel biomarkers for glycaemic deterioration in type 2 diabetes. The topic of the manuscript is potential interest. However, a series of specific issues, in particular regarding the experimental design and methods employed, somehow make the results presented not conclusive.

1. The reason why author choose these biomarkers (DCS, GoDARTS, ANDIS, GDF15, IL-18Ra, NogoR, CRELD1, FAS, ENPP7 et al) as the novel biomarkers to investigate.
2. When quantification of β -cell proliferation should double stain Edu and insulin, rather than Edu and pdx1.
3. The author should also detect the expression levels of apoptosis-related protein (such as bax, bcl2, Caspase et al) via Western blot.
4. Since author confirmed that NogoR affected β -cell apoptosis in vitro, the author should also test if NogoR affect β -cell apoptosis in vivo.
5. Author investigate the effect of IL-18Ra on IL-18Ra signalling should perform in the MIN6 cells not in the HEK293 cells.
6. Insulin content is never measured, and GSIS should have been measured per insulin content to compensate for changes in insulin production and size of the beta-cells. Is it even adjusted for number of islets? This is not clear.

7. The role of leucettine L41 used in the Figure 5d is completely missing in the manuscript. And author clarified that "None of the tested compounds affected human islet proliferation (Fig. 5d, line 512-513), while 10 uM leucettine L41 can enhance β -cell proliferation.

8. The information mice that used in vivo metabolic tests are not clear (such as age, weight). C56BL6J mice (Charles River) maintained for 4 weeks on a high fat high sucrose (HFHS) diet, the weight of mice has about 40g?

9. Line 507-508, the results of Figure 5d was performed in the human islets rather in the mouse islets.

Reviewer #3 (Remarks to the Author):

The study by Rutter et coll. provides a multi-omic analysis of several large European patient cohorts of existing type 2 diabetes with the purpose of identifying and validating biomarkers for disease progression in particular biomarkers associated with glycaemic deterioration post on-set of diabetes. In this collaborative effort within the European RHAPSODY initiative, targeted proteomics and metabolomics, untargeted lipidomics, and functional analysis of selected candidate markers were combined to both identify new biomarkers for disease progression and gain new insights into possible mechanism of disease on-set and progression. The study is robust, well-designed and the methods used are appropriate and data analysis and interpretation are appropriate. It is very appreciated that the authors used four independent patient cohorts to validate their biomarker findings in the discovery cohorts, increasing thus the confidence in the found biomarkers, which is such a pivotal aspect of the biomarker discovery, in general, but often underrepresented.

Therefore, I consider the work to be of high significance in the field of diabetes research and management, and to provide a solid multi-omic data reference for prospective studies, and hence suggest its publication after addressing the points below.

Major point:

The association of individual biomarker categories, i.e. omic data with glycaemic deterioration were primarily evaluated and discussed in the manuscript. However, the interrelation, correlation between the found biomarkers is not addressed. For example, a pathway analysis, interaction pathway analysis among and across lipid, metabolite and protein markers would strengthen the mechanistic understanding and the value of the study. In this context, for example, one of the protein biomarkers, ENPP7, which is in the intestine associated with ceramide and PC absorption, as well as SM hydrolysis, does not seem to render upon its level changes a correspondingly alteration in the plasma levels of this lipid categories. Or is the ENPP7 change reflected in /correlated with SM42:2;2? Such interaction between different categories of biomarkers and discussion thereof would expedite the understanding of the findings.

Minor points:

In Materials and Methods, please define the smallest time between diagnostic and sampling.

-What are the reference samples that were used for lipidomics analysis.

-For the in-vivo metabolic test as well as insulin secretion and human islets analysis please provide information on how many mice were used?

The line 462-464 needs rewording: it is not clear what the four and three refer to. Here also a brief explanation of what exactly the adjustments entails would improve clarity of the method.

-In conclusion the authors state that the novel biomarkers are suggestive of potentially distinct mechanisms of onset and progression of diabetes. However, the protein and lipid data reveal a similarity between the drivers of diabetes incidence and prevalence and progression. These statements need clarification, as they seem to be contradictory.

-Association between the content of dietary saturated and unsaturated fat with diabetes is a major, general topic of investigation. Could the authors comment on the possible effect of the (un)saturation level in TAG biomarkers on diabetes? They seem to have predominantly one double bond, which indicates a rather substantial amount of saturated fatty acids to be esterified. Can this relate to the inflammatory protein markers, and ultimately to the comorbid inflammatory conditions in diabetic patients?

Reviewer #4 (Remarks to the Author):

In this manuscript, Prof. Sliker et al. employ a metaanalysis of 3 discovery cohorts tested for metabolite, lipid, and protein biomarkers of development of an insulin requirement with T2D. The analysis showed significant association between homocitrulline (Hcit) and amino adipic acid (AADA) levels and diabetes progression, but external validation showed nonsignificant effects on incident diabetes. Multiple triglyceride species were also identified. The protein analysis showed significant HRs in 10 proteins, with similar effects on incident and prevalent T2D. The authors also performed Mendelian Randomization analysis of these proteins, which showed a likely causal relationship for 3 proteins- GDF15, IL18Ra and FAS. Finally, for two of the proteins (NogoR and IL-18RA), they go on to perform in vitro testing, as well as in vivo administration to mice to determine effects relevant to T2D. They found that NogoR appears to induce beta cell apoptosis at higher concentrations but may have an effect on insulin sensitivity that improves glucose tolerance when administered in vivo. Testing on IL18RA showed the ability to abrogate the impact of IL18 administration on NFKB activation in HEK cells, suggesting a beneficial effect on inflammation associated with T2D. In general the authors should be congratulated for taking this multidisciplinary approach that starts with a very large number of patient samples from several independent cohorts followed by translational experiments to understand the mechanisms of differences that they observe. However, I do think some changes could help the reader better interpret the relevance of some of the findings.

Major Comments

1. Multiple cohorts were used (which is a strength), but although these are all described by the authors, it gets a little confusing to interpret the results when reading the text, especially for a reader unfamiliar with the European cohorts and when combined with the abbreviations for the different metabolites, etc. I also got confused about why some analyses did and some did not include prevalent diabetes. It might be helpful to have a consort diagram or flow chart type figure with the different studies and their abbreviations, short bullets on populations they include, and the relevant analyses that were applied to them to help walk the reader through the analyses.
2. The study talks about biomarkers identified in terms of prediction of insulin requirement, incident diabetes or prevalent diabetes. The approach using hazard ratios makes sense, but I think it would also be helpful to see the actual values of the relevant biomarkers for each group so that the reader can understand the distribution and overlap between the groups and if the values were significantly different. They may be less helpful for prediction if there is substantial overlap.
3. Mendelian randomization analysis suggested that NogoR does not have a causal relationship with diabetes development and NogoR has differing effects depending on the concentration used (beta cell death vs improved in vivo glucose tolerance and insulin sensitivity). The authors conclude that the in vivo effect associated with lower circulating concentrations is likely the true effect. What were the concentrations in patients tested and how do these compare? Data from in vivo mouse concentrations are referred to in the discussion but not included in results that I saw. Were these in vivo levels of drug correlated with GTT or ITT results?

Minor Comments

4. Figure 1B and 1C- would make the labeling of the metabolites used a little clearer either in the figure or in the legend- 1b and 1c are not specifically called out in the text and it took me a bit and matching up the HRs to figure out what these represented.
5. Supplemental Figure 1- the baseline and insulin stimulated pAKT and AKT bands look very different for NogoR and CRELD. This seems unexpected since CRELD is being used for a control for impacts on insulin signaling? Also big differences in the loading control in the CRELD blot make it a little hard to interpret the changes in pAKT.
6. Table S2- some of the values for Lipidomics ANDIS discovery cohort seem like they are incorrect/were entered incorrectly (average BMI of 60?) diabetes duration of 809 years?

Reviewer #5 (Remarks to the Author):

Using three different cohorts consisting of ~3000 type 2 diabetes patients, the authors conducted multi-omics study of metabolomics, lipidomics and proteomics to identify biomarkers potentially useful for disease prognostics, disease mechanisms, and potential therapeutics to slow diabetes progression. Although the authors validated the results using data from four independent cohorts.

One of the important components of this paper is the statistical methodology for analyzing these complex high dimensional data which have complex data structures, (a) complex correlation structures between and within subjects and across data types, (b) potential missing values, (c) potential nonlinear relationships with covariates, and (d) high dimensionality in the data etc. The section on statistical analysis provides very limited description about the methods used. They provide a list of software packages along with Cox regression, logistic regression and ANOVA. All statistical methods and software make some assumptions regarding the data. For example, ANOVA assumes that the data are (at least asymptotically) normally distributed and more importantly, assumes that all groups have equal variance (homoscedasticity) for each outcome variable. These assumptions have not been verified. Increasingly, researchers are recognizing that metabolomics data are compositional. Are the methods used in this paper account for the compositionality of the data? The Cox regression model makes proportional hazard assumption. Is that valid? Given that the data are being integrated from multiple cohorts, there is a potential for heterogeneity. How is that accounted for? The Benjamini -Hochberg procedure is valid under some conditions. Are those conditions valid in the present analysis?

In summary, the section on Statistical Analysis must be expanded considerably to address various issues raised. Validity of the results of the paper hinge on the appropriateness of the statistical methods.

Reviewer #1 (Remarks to the Author):

Novel biomarkers for glycaemic deterioration in type 2 diabetes: an IMI RHAPSODY study

Sliker et. al., leverage several large European population cohorts to identify metabolomic, lipidomic, and proteomic biomarkers of T2D disease progression (defined by sustained insulin therapy or deterioration of HbA1c despite ≥ 2 non-insulin diabetes therapies). Five metabolites were nominally associated with the primary outcome after adjustments for age, sex, and BMI (n=2,975; n cases =612) including homocitrulline, amino adipic acid, isoleucine, glycocholic acid, and taurocholic acid. Only amino adipic acid and homocitrulline were significant after adjusting for multiple hypothesis testing. These associations were directionally consistent but not significant in the replication cohorts (n = 2,668; n cases =668). For lipids, 8 triglyceride species were associated with progression and SM 42:2;2 was protective (n = 2,608; n cases = 486). These findings were not validated, but all were positively associated with incident diabetes in MDS. Ninety-eight proteins were nominally associated with the primary outcome (n = 1,188; n cases = 271), of which MIC-1/GDF-15 replicated in 1 of the validation cohorts and NogoR replicated in the other replication cohort. Mendelian randomization supported a possible causal association of PE 18:0,0-18:2;0 and the aptamers GDF-15, IL-18Ra, and FAS with T2D. Functional studies for a select group of proteins (MIC-1/GDF-15, IL-18Ra, NogoR, CRELD1, FAS, and ENPP7) suggest these proteins do not affect insulin secretion in mice or human islet cells. IL-18Ra and NogoR increases apoptosis of mouse islet cells, but at super-physiologic doses. NogoR infusions over 2 weeks improved glucose tolerance in mice on HFHS chow and HepG2 cells incubated with NogoR had increased insulin-induced signaling phosphorylation products. IL-18Ra was also found to attenuate IL-18 signaling in a dual luciferase reporter cell assay. In summary, a number of metabolomic, lipidomic, and protein biomarkers were identified associated with diabetes progression, some of which may be causally associated to T2D development via MR analysis and preliminary functional studies suggest that NogoR may affect insulin signaling and IL-18Ra could exert its effects by suppressing IL-18 signaling.

A strength of this manuscript is the leveraging of omics technology to identify biomarkers of T2D progression—which is an important clinical question that has not been previously studied in such large numbers of individuals. They also followed up their biomarker findings with preliminary functional studies to begin to elucidate biological mechanisms. Limitations are that only a few of these biomarkers actually replicate for the primary endpoint despite relatively large numbers and cases in the validation cohorts which diminishes the impact of these findings. Secondly, the primary outcome that was studied was diabetes progression while the functional studies were conducted in euglycemic mice (albeit fed a HFHS diet) and cells that had normal glucose homeostasis and insulin signaling. Thus, conclusions are more related to how these proteins potentially contribute to T2D development rather than their contribution to disease progression which is arguably the main focus of the paper.

>We thank the referee for these insightful and helpful comments. As elaborated in detail below, in addition to studies in mice with only mild glucose intolerance (HFD) we now include data using a more severe model of hyperglycemia, the *db/db* mouse. As suspected by the referee, the impact of NogoR appears to be different in this versus the less metabolically abnormal model.

Major critiques:

1. In the Molecular Measurements section, it states sample collection occurred between 0-3 years of diagnosis. What is the mean/median years between time of diagnosis to the lab draw? Were any

sensitivity analyses conducted, especially if there was significant variability in this time, to determine how this could influence the primary analyses?

>The reviewer makes an important point. For the discovery metabolomics set the median diabetes duration was for DCS 2.6 years, for GoDARTS 1.4 years and for ANDIS 0 years. In case of the latter cohort, blood samples were taken at diagnosis. For the lipidomics set the median in DCS was 2.0 years and 1.4 years in GoDARTS and 0 in ANDIS. Finally, in the proteomics set the median in DCS was 1.5 years and 0.92 in GoDARTS. Furthermore, as a sensitivity analysis we adjusted model 1 (age, sex, BMI) additionally for diabetes duration in each cohort and meta-analysed across cohorts. Based on this analysis, the effect of diabetes duration was marginal (see figure below). The Pearson correlation for the hazards of the metabolites before and after this additional adjustment was 0.96 (95%CI:0.89-0.98), for lipids 0.95 (95%CI:0.93-0.96) and for proteins 0.91 (95%CI:0.91-0.93).

We have added this to the methods (**Page 21, Lines 515-518**):

The effect of diabetes duration was also investigated on itself, but this did not influence the results, with high correlation between effect sizes before and after adjustment: metabolites 0.96 (95%CI:0.89-0.98), lipids 0.95 (95%CI:0.93-0.96) and proteins 0.91 (95%CI:0.91-0.93).

2. Fasting plasma glucose and HbA1c are known predictors of T2D progression, how strongly are the identified biomarkers of progression correlated to these clinical measures? Are there effects attenuated after adjustments for these values?

>The reviewer is completely right and we apologize that this was not clear from the manuscript. All models were stratified for HbA1c levels (<53 mmol/mol, 53-75 mmol/mol and >75 mmol/mol) to account for this effect. We describe this now in the methods (**Page 21, Lines 514-515**):

All three models were stratified for HbA1c (strata: < 53 mmol/mol, 53-75 mmol/mol and >75 mmol/mol).

3. It is increasingly acknowledged that there are subtypes of T2D that are associated with different rates of progression to insulin therapy and complications. Have you considered looking at if BMI and c-peptide levels are effect modifiers of these associations with T2D progression?

>This is an interesting suggestion. Note that, in the base model, we already adjust for BMI. To investigate a possible interaction between BMI and C-peptide we performed an interaction model between BMI/ C-peptide and the top 25 identified biomarkers shown in Figures 1-3. For BMI, only isoleucine and homocitrulline showed a modest interaction. These data are summarized below for

the reviewer's information only. The tables show the hazard ratio for the interaction term between the biomarker and BMI or C-peptide.

Variable	HR	Lower	Upper	Zval	Pval	I2	Het
Ile	1.02	1.00	1.03	2.35	0.02	0.00	0.22
Hcit	1.02	1.00	1.04	2.18	0.03	0.00	0.23

>For C-peptide, there were also two biomarkers that showed a modest interaction, which were HEMK2 and aminoadipic acid.

Variable	HR	Lower	Upper	Zval	Pval	I2	Het
HEMK2	0.83	0.72	0.96	-2.52	0.01	0.00	0.92
AADA	1.11	1.02	1.20	2.48	0.01	0.00	0.56

>This suggests that for the top biomarkers the effect of BMI relatively limited, while the effect of C-peptide was larger for the top three biomarkers suggestive of a relation between these biomarkers and C-peptide.

We have also added this to the results.

Page 23-24, Lines 562-564:

Furthermore, homocitrulline showed an modest interaction with BMI (P=0.03) which could to some extent mask the differences in levels at baseline. For AADA, however an interaction with C-peptide was observed (P=0.01).

Page 24, Lines 574-575:

In addition, isoleucine levels showed an modest interaction with BMI (P=0.02).

Page 25, Lines 611-612:

Levels of HEMK2 showed an interaction with C-peptide levels (P=0.01).

4. Incident and prevalent diabetes associations were reported for some but not all the metabolites discussed in the results section, what were the associations for the other metabolites or were they not measured in MDC and DESIR?

>The reviewer is correct that the metabolites were not available in these other cohorts. We have now made this clearer by including a flow chart as Supplemental Figure 1 and is this figure is shown below.

Supp. Fig. 1

5. In lipidomics a frequent challenge is successful separation of related lipid species (i.e., different TAG species often cluster together making it difficult to differentiate between those with similar acyl chain lengths). Given the top TAG findings all have similar acyl carbon lengths, how confident are you of the precision of these lipid measurements? Also given there were 3 cohorts with available lipidomics data, could one of them have been used for validation of the primary findings?

>We recognize the importance of the reviewer's comment and have added the mean and SD for each of the TAGs in **Table S2** and added the following section to the methods (**Page 10, Lines 232-240**):

Validation of identity between cohorts was achieved by confirming that each TAG species was identified with a similar fatty acid profile. Spectra were analysed with in-house developed lipid identification software based on LipidXplorer. 20 TAGs are quantified as species (e.g. TAG 48:1;0). Fatty acid amounts within TAG species were calculated based on intensities of neutral losses of fatty acid fragments. Only fatty acids that were measured in at least one cohort in 80% of the subjects were considered. Profiles were standardized on each species such that fatty acid amounts within one species sum up to 100% within every subject. From this data mean and standard deviations across each cohort were calculated (Table S3).

And the head of Table S3:

species	FA	mean_DCS	sd_DCS	mean_GO_DARTS	sd_GO_DARTS	mean_ANDIS	sd_ANDIS
TAG							
46:1;0	12:0;0	14.74986102	3.629979992	15.23397188	4.112580703	14.96555443	3.619386042
TAG							
46:1;0	14:0;0	22.98545009	5.18025807	20.99394572	4.44875846	24.24228191	6.621795238
TAG							
46:1;0	16:0;0	30.62684454	3.659455032	31.0229173	2.61603218	32.16165643	5.721503037
TAG							
46:1;0	16:1;0	12.1548948	3.969676084	11.83658161	4.472316622	13.7753373	7.212793774
...

6. For the proteomics validation please show the meta-analysis results for the ACCELERATE and ANDIS trial in Figure 3.

>We thank the reviewer for this suggestion. We have now added this to Figure 3 (update figure also shown below):

Fig. 3

Updated Figure 3.

Also, were HSP 908, SMAC, Coactosin-like protein, Testican-1, and HEMK2 not measured in either validation cohorts?

>Due to limited resources we had to decide to validate only the top protein biomarkers that were identified as a risk factor and not those with a lower risk / protective effect. However, we entirely agree that these biomarkers will be interesting to investigate in more detail in future studies.

7. As alluded to in the limitations, specificity of aptamer measurements in the SomaLogic screen is always a concern. However, sincere genetic data is available, were the reported aptamers associated with disease progression supported by GWAS hits that mapped to pQTLs?

>While this is an interesting suggestion, this would require a GWAS on time to insulin. To the best of our knowledge, no GWASs have been reported on diabetes progression, to date. We have considered running a GWAS on diabetes progression, for example on time to insulin initiation, but this would be severely underpowered.

8. Given the focus of the paper on biomarkers of diabetes progression, were any biomarkers associated with progression that were not associated with incident diabetes?

>This is indeed an interesting question. However, as can be appreciated from the various validations, biomarkers were generally both a biomarker for incident diabetes as well as diabetes progression. SM-

42:2;2 was the exception where there was an opposite effect for incident diabetes and diabetes progression.

9. NogoR was shown to cause beta cell apoptosis at likely supra-physiologic doses but in healthy beta cells. It would be interesting to see NogoR exposure in beta cell models of diabetes.

>This is an interesting suggestion. As also discussed in response to referee # 4 point 3, we attempted to explore the effects of NogoR in the presence of agents (high levels of fatty acids or of cytotoxic cytokines; the ER stressor tunicamycin) which partly mimic the effects of type 2 diabetes. We did not observe any clear potentiation of the effects of NogoR (nor of other tested biomarkers) of these agents on apoptosis measured at 48 in mouse islets (shown below for the reviewer's information)

10. Arguably, long term exposure to elevated levels of metabolites and proteins associated with incident disease and disease progression are what cause these outcomes. Have efforts been made to study animal models of more chronic exposures to elevated levels of NogoR?

>We thank the reviewer for this suggestion. We have now performed a new set of experiments with a further cohort of mice, specifically the *db/db* model of hyperphagia and obesity-induced diabetes. In these experiments we treated the animals for four weeks, rather than two, using osmotic minipumps to achieve steady delivery of NogoR for the entire period. The new data are shown in the

revised Figure 6h-p. Interestingly, in this model of more severe hyperglycemia (fasting glycemia > 15 mM) *versus* high fat diet alone (Fig 6 a-g; FG < 7 mM), the effects of NogoR infusion calculated to provide a similar dose to that achieved by daily injection were more modest, but we observed a tendency towards *impaired* insulin sensitivity (Fig. 6n), and Results. Hence, the impact of NogoR in man is likely to be dependent upon disease severity, as well as the extent and duration of NogoR exposure. Future studies, with alternative models of even more severe insulinopenic diabetes in which complications (retinopathy, liver failure, retinopathy etc) are observed (e.g. streptozotocin treatment), will be needed to explore these changes but are beyond the scope of the present report.

The results are described on **Page 27-28 in Lines 662-667**):

In contrast, when introduced chronically into the more severely diabetic db/db mouse model, NogoR had no discernible effect on oral glucose tolerance (Fig 6h-i) or insulin secretion (Fig 6j,m) but tended to increase fasting blood glucose and to impair insulin sensitivity (Fig 6n,o), in line with the expected increase in circulating NogoR levels (Fig. 6p). Beta cell mass was not significantly affected by NogoR (6.1 ± 3.2 , $n=5$ and 9.1 ± 4.2 , $n=4$, for NogoR-treated and control db/db mice, respectively, $P=0.264$).

11. Based on the functional study results, IL18-Ra should be protective for diabetes and disease progression given it blocks the inflammatory effects of IL-18. However, this is opposite of positive association found in the population cohorts. What is the proposed reason for this discrepancy in findings?

>We do not agree that these findings represent a discrepancy. Rather, we would question whether the direction of effect can be predicted based on the association between IL18Ra levels and progression rate measured. It could equally be – as our data tend to suggest – that IL18Ra exerts a protective effect, and that this protein provides an excellent biomarker for disease progression – as our data indicate – but do not contribute positively (i.e. to accelerate) the disease process. In any case, we would note that high levels of IL18Ra actually increased apoptosis slightly in mouse and human islets, through mechanisms which are currently unclear, but may involve a complex interaction with multiple receptors on the islet. We would stress whilst IL-18 leads to the production of nitric oxide and apoptosis in mouse islets (Lewis, PNAS 2016), deletion of the IL-18 receptor accelerated graft failure, demonstrating an essential role for signalling by IL-18R to maintain islet viability. Other data suggest that IL-18Ra might interact directly with single immunoglobulin-IL-1-related receptor (SIGIRR, also called IL-1R8 and TIR8) which in turn inhibits anti-inflammatory signalling by IL-37 (Wald, Nat. Immunol, 2003). The current data (Fig. 5a) indicate that recombinant IL-18Ra leads to a mild increase in apoptosis in mouse islets (a tendency in this direction is also seen in human islets; Fig. 5c), consistent with a direct effect of IL-18Ra on this or another non-canonical receptor present in on islet cells. Taken together, these findings suggest that sequestration of IL-18 by IL-18Ra, as well as direct signalling by IL-18Ra, may affect β cell function and/or survival, and hence disease progression.

These points are now made in the revised manuscript **Page 32, Lines 782-783**.

For IL18Ra it is unclear how its increased plasma levels contribute positively to the disease progression process.

Page 33, Lines 792-794:

Other data suggest that IL-18Ra might interact directly with single immunoglobulin-IL-1-related receptor (SIGIRR, also called IL-1R8 and TIR8) which in turn inhibits anti-inflammatory signalling by IL-37.⁶⁸

Page 33, Lines 798-801:

In a previous study it was shown that IL-18 leads to the production of nitric oxide and apoptosis in mouse islets, while deletion of the IL-18 receptor accelerated graft failure, demonstrating an essential role for signalling by IL-18R to maintain islet viability.⁶⁵

Page 33, Lines 804-806:

Taken together, these findings suggest that sequestration of IL-18 by IL-18Ra, as well as direct signalling by IL-18Ra, may affect β cell function and/or survival, and hence disease progression.

12. In the conclusions section, given the functional studies were all conducted in animal and cell models with normal glycemia it is premature to conclude that these findings provide “new mechanistic insights” into glycemic deterioration after the development of diabetes. This section should be revised to reflect more accurately what was actually studied.

>We agree, and although our studies are now extended to a model of frank hyperglycemia (db/db mice) we have tempered these statements accordingly (Discussion, **Page 35, lines 850-852**)
Our findings highlight new molecular changes that associate with glycaemic deterioration once diabetes has developed.

Minor critiques:

1. There are discrepancies in aptamer nomenclature throughout the manuscript that should be corrected (ex. GDF15 which is also MIC-1 is referred as GDF15 in the text, GDF15/MIC-1 in Fig 3, and MIC-1 in Supplemental T6).

>We have now modified all instances of MIC-1 to GDF15 and mention only once that this protein has two names.

2. In Figure 4, what is the difference between 4a and 4b?

>In Fig. 4a, glucose was 3 mM, and in Fig. 4b, 17 mM, as indicated by the “Vehicle” label in each case.

3. In the limitations section (page 28, line 676) “A third limitation...” should actually be a second limitation

>We apologize for the oversight and have changed this to “second”.

Reviewer #2 (Remarks to the Author):

Comments:

Remarks to the Author

In this manuscript, Roderick C Sliker et al use many diabetes individuals to investigate novel biomarkers for glycaemic deterioration in type 2 diabetes. The topic of the manuscript is potential interest. However, a series of specific issues, in particular regarding the experimental design and methods employed, somehow make the results presented not conclusive.

1. The reason why author choose these biomarkers (DCS, GoDARTS, ANDIS, GDF15, IL-18Ra, NogoR, CRELD1, FAS, ENPP7 et al) as the novel biomarkers to investigate.

>We describe this on **page 26, Line 636-639**:

We chose to study six protein biomarkers with greatest effect size (GDF15, IL-18Ra, NogoR, CRELD1, FAS, ENPP7) and which accelerated progression, i.e., those which may plausibly exert a deleterious effect on insulin secretion or action.

2. When quantification of β -cell proliferation should double stain Edu and insulin, rather than Edu and pdx1.

>We appreciate the reviewers' suggestion. However, we note however that insulin is in the medium used in these studies and therefore staining for insulin might result in false positive staining for insulin (after endocytosis). Therefore, and to avoid this risk, we chose to stain for Edu/Pdx1 which are both nuclear.

3. The author should also detect the expression levels of apoptosis-related protein (such as bax, bcl2, Caspase et al) via Western blot.

>Thank you for this excellent suggestion. We attempted to detect a change in BCL2, BCL-XBID, BAX and HIAPP protein levels in human islets in response to NogoR but were unable to detect any such change with the currently available antibodies to these proteins.

An example blot for BCL-X is shown below, for the reviewer's inspection.

Effect of NogoR (100 nM) on BCL-X protein levels in human islets. Groups of human islets (around 400 islets per group) from three human healthy donors were cultured in the regular islet culture media with 100nM NogoR protein for 48h. Islets were harvested in the RIPA buffer supplemented with protease inhibitors. Protein lysates were applied to SDS-PAGE, proteins transferred to PVDF membranes and probed with antibodies against the BCL-2 related proteins and beta-actin.

As an alternative approach to validating our results obtained using the Caspase kit we therefore performed TUNEL assays. These data are now added as a new supplementary figure (**SFig.7**), below

Supp. Fig. 7

Supplemental Figure 7. The TUNEL staining of human pancreatic islets treated with cytokines and recombinant human NogoR protein. β -cell apoptosis was analyzed by staining of cultured human pancreatic islets of TUNEL, insulin, glucagon and DAPI. Representative islet images treated with vehicle (A), cytokines (B) and NogoR (C). (D) The percentage of TUNEL positive β - cells was calculated in islets

from 3 different human islet donors. Results shown are means \pm SEM. ** $p < 0.01$ *** $p < 0.001$ versus the vehicle group.

Results (Page 27, Lines 649-652):

The increased apoptosis was further illustrated based on a TUNEL assay which measures apoptotic DNA fragmentation. A significant increase (SFig. 7A-D) in percentage of TUNEL positive nuclei – indicative of increased apoptosis – was observed in cells exposed to NogoR (8.5%) versus vehicle (3.7%).

Methods (Page 20, Lines 487-498):

TUNEL staining of human pancreatic islets.

Human pancreatic islets from 3 independent donors were cultured for 72h with the recombinant cytokine mix or recombinant NogoR protein at concentrations used for the caspase assay. Islets were pelleted into Histogel (Thermo Scientific), processed and embedded into paraffin. Paraffin blocks were cut into sequential sections 4 μ m apart. To maximize the number of islets stained, 3 non-sequential slides chosen from each block for staining and analysis. Slides were stained with antibodies for insulin (A0564, Agilent Dako, Santa Clara, CA), glucagon (PU039-UP, Biogenex, Fremont, CA) and with DAPI (Thermo Scientific). Cell death was determined with the TUNEL assay using ApopTag In Situ Apoptosis Detection Kit (EMD Millipore). Slides were scanned with the PANNORAMIC 1000 scanner (3DHistech, Budapest, Hungary) and images were processed with Visiopharm software (Westminster, CO) to determine % of TUNEL insulin positive nuclei.

4. Since author confirmed that NogoR affected β -cell apoptosis in vitro, the author should also test if NogoR affect β -cell apoptosis in vivo.

>The referee makes an interesting suggestion. We have now measured beta cell mass in db/db mice treated with NogoR. Beta cell mass in the saline group was 9.05(SD=4.22, N=4) and in the NogoR group 6.05 (SD=3.22, N=5) which suggests a tendency towards lower beta cell mass although the difference was insignificant (P=0.265). Given the likely scarcity of apoptotic beta cells in both control and NogoR-treated animals, no attempt was made to measure the latter directly. We do not feel that measurements of apoptosis on these sample would be likely to yield any measurable results.

We have added this to the text in the following (Page 27, Lines 662-667):

In contrast, when introduced chronically into the more severely diabetic db/db mouse model, NogoR had no discernible effect on oral glucose tolerance (Fig 6h-i) or insulin secretion (Fig 6j,m) but tended to increase fasting blood glucose and to impair insulin sensitivity (Fig 6n,o), in line with the expected increase in circulating NogoR levels (Fig. 6p). Beta cell mass was not significantly affected by NogoR (6.1 ± 3.2 , n=5 and 9.1 ± 4.2 , n=4, for NogoR-treated and control db/db mice, respectively, P=0.264).

5. Author investigate the effect of IL-18Ra on IL-18Ra signalling should perform in the MIN6 cells not in the HEK293 cells.

>While ideally one would test the signalling on beta or beta- like cells such as MIN6, we would note that MIN6 cells don't naturally express the canonical IL18 receptor which complicates this decision. The engineered HEK293 cells deployed, on the other hand, overexpress the canonical receptor alongside a reporter and therefore provide a useful model system.

As discussed in response to referee #1, point 11, it is conceivable that the actions of IL-18Ra on the islet (notably an increase in apoptosis) are achieved through an action on non-canonical IL-18Ra receptor, perhaps influencing signalling through IL-17. Whilst these are all exciting possibilities we

feel that the studies required to explore these interactions lie beyond the scope of the already complex manuscript. We added this notion to the discussion in **Page 33, Lines 792-794**:

Other data suggest that IL-18Ra might interact directly with single immunoglobulin-IL-1-related receptor (SIGIRR, also called IL-1R8 and TIR8) which in turn inhibits anti-inflammatory signalling by IL-37.⁶⁸

6. Insulin content is never measured, and GSIS should have been measured per insulin content to compensate for changes in insulin production and size of the beta-cells.

>The reviewer makes a fair point. However, we would like to point out that these incubations were performed over 1h only, hence stimulus-dependent changes in total islet insulin content are likely to be small or zero. Indeed, correction of total insulin content is sometimes used in batch incubations such as these to avoid variability due to differences between islet number or size between individual incubations. As indicated by the rather small excursions from the mean of individual data points, this was not an issue in the present studies.

Is it even adjusted for number of islets? This is not clear.

>The same number of islets was used in each case. This is now indicated in the figure legend of Figure 4:

Figure 4. Impact of identified biomarkers on insulin secretion from mouse (a,b) and human (c) islets. Incubations were performed for 30 min. at the indicated concentrations of glucose, and secreted insulin measured using an electrochemiluminescence assay. ***, ** p = 0.001, 0.01 compared to vehicle; ##p<0.01 vs 3 mM glucose. Comparisons by one-way ANOVA in each case. The number of islets was the same across experiments. Other details are given in the Methods Section.

7. The role of leucettine L41 used in the Figure 5d is completely missing in the manuscript. And author clarified that "None of the tested compounds affected human islet proliferation (Fig. 5d, line 512-513), while 10 uM leucettine L41 can enhance β -cell proliferation.

>We apologize for the confusion. Leucettine L41 was used as a positive control in this experiment. We have now made this clear in the legend to Figure 5 (Line 1170):
The DYRK1A/DYRK2/CLK kinase inhibitor leucettine L4179 was used as a positive control.

Methods (Page 20, Line 469-471):

After overnight recovery, mouse or human islets were cultured for 72h in 12-well plates (200-300 islets per well) in RPMI-1640 medium containing 5mM glucose, 2% FBS, 10 μ M EdU and tested proteins. Leucettine L41 (10 μ M) was used as a positive control.

8. The information mice that used in vivo metabolic tests are not clear (such as age, weight). C56BL6J mice (Charles River) maintained for 4 weeks on a high fat high sucrose (HFHS) diet, the weight of mice has about 40g?

>We apologize if this was unclear, but please note that the body weight of the mice used in the metabolic tests are given in Figure 6a,c,f (also below).

9. Line 507-508, the results of Figure 5d was performed in the human islets rather in the mouse islets.

>We are not sure what the problem may be here. We refer only to human islets in the text.

Reviewer #3 (Remarks to the Author):

The study by Rutter et coll. provides a multi-omic analysis of several large European patient cohorts of existing type 2 diabetes with the purpose of identifying and validating biomarkers for disease progression in particular biomarkers associated with glycaemic deterioration post on-set of diabetes. In this collaborative effort within the European RHAPSODY initiative, targeted proteomics and metabolomics, untargeted lipidomics, and functional analysis of selected candidate markers were combined to both identify new biomarkers for disease progression and gain new insights into possible mechanism of disease on-set and progression. The study is robust, well-designed and the methods used are appropriate and data analysis and interpretation are appropriate. It is very appreciated that the authors used four independent patient cohorts to validate their biomarker findings in the discovery cohorts, increasing thus the confidence in the found biomarkers, which is such a pivotal aspect of the biomarker discovery, in general, but often underrepresented. Therefore, I consider the work to be of high significance in the field of diabetes research and management, and to provide a solid multi-omic data reference for prospective studies, and hence suggest its publication after addressing the points below.

Major point:

The association of individual biomarker categories, i.e. omic data with glycaemic deterioration were primarily evaluated and discussed in the manuscript. However, the interrelation, correlation between the found biomarkers is not addressed. For example, a pathway analysis, interaction pathway analysis among and across lipid, metabolite and protein markers would strengthen the mechanistic understanding and the value of the study.

>While we appreciate the reviewer's comment, the number of identified biomarkers is relatively low, making meaningful enrichment analysis challenging. For example, neither enrichment of the FDR significant lipids and those with a nominal P-value < 0.05 reached the threshold for significant pathway enrichment based on the LIPEA tool (<https://lipea.biotech.tu-dresden.de/>). The same applies to the proteins, where we did not find a significant enrichment in any annotation for the top proteins with all measured proteins as the background set (StringDB).

In this context, for example, one of the protein biomarkers, ENPP7, which is in the intestine associated with ceramide and PC absorption, as well as SM hydrolysis, does not seem to render upon its level changes a correspondingly alteration in the plasma levels of this lipid categories. Or is the ENPP7 change reflected in /correlated with SM42:2;2? Such interaction between different categories of biomarkers and discussion thereof would expedite the understanding of the findings.

>We agree with the reviewer that looking at the correlation among identified biomarkers may be interesting to identify a potential relation between the proteins, lipids and metabolites. We have therefore now investigated this correlation in the three cohorts (proteins only in DCS/GoDARTS). As can be appreciated from the figure below, there is particularly a strong correlation between the TAGs, but less so for the proteins and metabolites. Clearly, protective biomarkers were negatively correlated to those that were risk factors as one would expect. ENPP7 protein levels were indeed correlated with lipids, but more with the TAGs than with SM42:2;2.

We have added this finding to the Results (Page 25, Line 597):

Nonetheless, the levels of TAGs were strongly correlated among each other (SFig. 5).

And results (Page 25-26, Lines 613-616):

Levels of testican-1 and HEMK, SMAC, coactosin-like protein were correlated (SFig. 5). At baseline not all proteins showed a clear upregulation in level in incident insulin users; the most profound effects were observed for NogoR, IL-18 Ra, ENPP7, HSP 90b (SFig. 6).

Minor points:

In Materials and Methods, please define the smallest time between diagnostic and sampling.

>The smallest time between the diagnosis and sampling were those individuals where the diagnosis and sampling was on the same date and this was mainly the case for ANDIS (t=0). Also see Table S1.

-What are the reference samples that were used for lipidomics analysis.

>We apologize that this information was omitted. The reference samples are aliquots of the same pool human plasma which have been purchased in large quantities. These aliquots are all identical and can therefore be used as an internal standard. We have added this to the methods (**Page 10, Lines 242**):
The reference samples are replicates from a pool of purchased plasma.

-For the in-vivo metabolic test as well as insulin secretion and human islets analysis please provide information on how many mice were used?

>This information is given in the figures and indicated by the number of data points in each bar in histograms

The line 462-464 needs rewording: it is not clear what the four and three refer to. Here also a brief explanation of what exactly the adjustments entails would improve clarity of the method.

>We have updated the Methods section regarding the adjustments performed and rephrased the sentence the reviewer is referring to.

Results (**Page 24, Lines 586-589**):

Further adjustment in the discovery cohort attenuated the effect size but the direction remained the same. Furthermore, in the partly (HDL, C-peptide) and fully adjusted model (additional adjustment for diabetes duration, glucose lowering drugs) four and three lipids remained significant, respectively (Table S6).

We also describe the models in the methods section (**Page 21, Lines 506-512**):

In each of the three cohorts DCS, GoDARTS and ANDIS, three Cox proportional hazard models were performed:

Model 1: Biomarker, age, sex, BMI

Model 2: Biomarker, age, sex, BMI, HDL, C-peptide

Model 3: Biomarker, age, sex, BMI, HDL, C-peptide, diabetes duration, glucose-lowering drugs

-In conclusion the authors state that the novel biomarkers are suggestive of potentially distinct mechanisms of onset and progression of diabetes. However, the protein and lipid data reveal similarity between the drivers of diabetes incidence and prevalence and progression. These statements need clarification, as they seem to be contradictory.

>We agree with the reviewer that this statement is not a logical conclusion from the Results section. Therefore we have omitted this statement and rephrased the sentence to (**Page 35, Lines 851-852**):

Importantly, we describe novel biomarkers for type 2 diabetes progression of different chemical classes.

-Association between the content of dietary saturated and unsaturated fat with diabetes is a major, general topic of investigation. Could the authors comment on the possible effect of the (un)saturation level in TAG biomarkers on diabetes? They seem to have predominantly one double bond, which indicates rather substantial amount of saturated fatty acids to be esterified. Can this relate to the

inflammatory protein markers, and ultimately to the comorbid inflammatory conditions in diabetic patients?

>The reviewer raises an important point. In our previous work (Slieker et al. Diabetes 2021) based on a previous paper (Rhee, JCI, 2011) we have shown that the obese diabetes subtype has higher levels of TAGs with shorter chains and lower double bonds. We have now performed the same analysis in the current study. We observe a similar pattern: high levels of TAGs with short chain length and lower number of double bonds are associated with higher risks of diabetes progression.

We have added these results to the methods and supplemental figures (**Page 25, Lines 592-596**):

As observed previously by us³¹ based on a previous report²⁹, TAG acyl chain length and number of double bonds determined the magnitude of effect of TAGs. In this study we also observe an almost linear relation between the acyl chain length and the number of double bonds and the hazard ratio, where the highest hazard was observed for the TAGs with the shortest acyl chains and the lowest number of double bonds (SFig. 4).

Reviewer #4 (Remarks to the Author):

In this manuscript, Prof. Sliker et al. employ a metaanalysis of 3 discovery cohorts tested for metabolite, lipid, and protein biomarkers of development of an insulin requirement with T2D. The analysis showed significant association between homocitrulline (Hcit) and amino adipic acid (AADA) levels and diabetes progression, but external validation showed nonsignificant effects on incident diabetes. Multiple triglyceride species were also identified. The protein analysis showed significant HRs in 10 proteins, with similar effects on incident and prevalent T2D. The authors also performed Mendelian Randomization analysis of these proteins, which showed a likely causal relationship for 3 proteins- GDF15, IL18Ra and FAS. Finally, for two of the proteins (NogoR and IL-18RA), they go on to perform in vitro testing, as well as in vivo administration to mice to determine effects relevant to T2D. They found that NogoR appears to induce beta cell apoptosis at higher concentrations but may have an effect on insulin sensitivity that improves glucose tolerance when administered in vivo. Testing on IL18RA showed the ability to abrogate the impact of IL18 administration on NFKB activation in HEK cells, suggesting a beneficial effect on inflammation associated with T2D. In general the authors should be congratulated for taking this multidisciplinary approach that starts with a very large number of patient samples from several independent cohorts followed by translational experiments to understand the mechanisms of differences that they observe. However, I do think some changes could help the reader better interpret the relevance of some of the findings.

>We thank the reviewer for these positive and helpful comments.

Major Comments

1. Multiple cohorts were used (which is a strength), but although these are all described by the authors, it gets a little confusing to interpret the results when reading the text, especially for a reader unfamiliar with the European cohorts and when combined with the abbreviations for the different metabolites, etc. I also got confused about why some analyses did and some did not include prevalent diabetes. It might be helpful to have a consort diagram or flow chart type figure with the different studies and their abbreviations, short bullets on populations they include, and the relevant analyses that were applied to them to help walk the reader through the analyses.

>We thank the reviewer for this suggestion. We have now added a supplemental figure (SFig 1) with a flow chart describing the current study, the cohorts and analyses performed. We hope this provides additional insight into our study design, and the cohorts involved.

2. The study talks about biomarkers identified in terms of prediction of insulin requirement, incident diabetes or prevalent diabetes. The approach using hazard ratios makes sense, but I think it would also be helpful to see the actual values of the relevant biomarkers for each group so that the reader can understand the distribution and overlap between the groups and if the values were significantly different. They may be less helpful for prediction if there is substantial overlap.

>The reviewer raises an important point. We have now included the figures showing the values for the top biomarkers in supplemental figures (also show below). We did observe that for the majority of the biomarkers there was a clear increase or decrease in biomarker levels depending on the direction of effect. For example, for the metabolites four out of the five metabolites showed increased levels in incident insulin users versus non-users. The levels of homocitrulline were however equal in both groups. For the lipids, the levels were all in the expected direction, with higher levels of TAGs in

the incident insulin group and lower in the non-insulin users, while for SM 42;2:0 opposite effects were observed. A similar observation was seen for the proteins, although for GDF15 a difference was not observed at baseline or even an opposite effect. It should be noted that GDF15 is known to be associated with other covariates such as age – for which we adjust - which could distort this figure. On the other hand, for clinical applications this makes GDF15 a less obvious candidate. We include this in the following sections:

Results (Page 23, Lines 560-562):

Of note, for AADA higher levels were observed at baseline for incident insulin users versus non-insulin users, but not for homocitrulline (SFig. 2).

Results (Page 25, Lines 590-592):

In line with the protective hazard ratio, the levels of SM 42;2;2 were lower in the incident insulin users versus non-insulin users (SFig. 3).

Results (Page 25-26, Lines 613-616):

Levels of testican-1 and HEMK, SMAC, coactosin-like protein were correlated (SFig. 5). At baseline not all proteins showed a clear upregulation in level in incident insulin users; the most profound effects were observed for NogoR, IL-18 Ra, ENPP7, HSP 90b (SFig. 6).

3. Mendelian randomization analysis suggested that NogoR does not have a causal relationship with diabetes development and NogoR has differing effects depending on the concentration used (beta cell death vs improved in vivo glucose tolerance and insulin sensitivity). The authors conclude that the in vivo effect associated with lower circulating concentrations is likely the true effect. What were the concentrations in patients tested and how do these compare?

>The referee raises a very interesting point. Given the apparently deleterious effects of NogoR on islet survival in vitro (Fig. 5) but protection against the development of glucose intolerance in vivo (Fig. 6a-g), we wondered whether this may reflect different actions of different concentrations of this protein. Thus, in vitro, we observed deleterious effects of NogoR to promote islet cell apoptosis only at concentrations only ≥ 3 nM (~ 120 ng/mL). In contrast, in vivo concentrations in the mouse are only ~ 25 pM (1 ng/mL). In humans, the majority of subjects have plasma NogoR levels of 2 ng/mL or below, with a tail of 70 individuals with concentrations >5 ng/mL up >90 ng/mL (2.2 nM), i.e. at levels potentially capable of prompting apoptosis, at least over the time frames used here.

Density plot of the NogoR concentrations measured in ACCELERATE.

These data would seem to make it unlikely that a deleterious action of NogoR on islet survival pertains *in vivo*. Nevertheless, and mentioned above, we explored the possibility that there may be interactions between NogoR and other circulating factors, that might potentiate the effects of this protein on apoptosis, notably fatty acids and selected cytotoxic cytokines. However, no potentiation of the effects of NogoR on islet apoptosis was observed in the presence of these agents (not shown).

We also explored the possibility that the relationship between plasma NogoR levels in subjects, and diabetes progression, may be biphasic i.e. low and high levels accelerating progression, but not intermediate levels. This relation was possibly present in ACCELERATE, but not in ANDIS where NogoR levels were assessed using the same technique.

Figure of the log levels of NogoR levels vs the hazard ratio based on a spline with two degrees of freedom in ACCELERATE.

Figure of the log levels of NogoR levels vs the hazard ratio based on a spline with two degrees of freedom in ANDIS.

These observations hint at such a biphasic relationship between NogoR levels and rate of disease progression, and thus possibly suggest that NogoR acts on insulin-sensitive tissues to potentiate insulin action (given that only negative actions, i.e. apoptosis, of the protein were observed on islets). Whilst our preliminary studies hint at this possibility (**SFig. 1**, now **SFig. 8**) the effects of NogoR on proximal insulin signalling in the human hepatocyte line HepG2 did not reach statistical significance. Extensive further studies using primary cells (hepatocytes, adipocytes, myotubes etc.) as well as more detailed metabolic studies *in vivo* in mice (euglycemic-hyperinsulinemic clamps using radiotracers) will be needed to dissect the actions of this protein on the relevant tissues. Such studies are expensive and likely to take 12-18 months and we therefore believed to lie outside the scope of the present report.

Data from *in vivo* mouse concentrations are referred to in the discussion but not included in results that I saw. Were these *in vivo* levels of drug correlated with GTT or ITT results?

>Thank you. In our new study in *db/db* mice we performed measurements of plasma NogoR at the end of the study (**Fig 6P**). Note that only a marginal difference between the injected and control groups at this time point and likely reflects gradual degradation of the compound over the course of the experiment. Exploring correlations between NogoR and IPGTT was not possible and in any case would be expected to be underpowered to detect meaningful changes.

Minor Comments

4. Figure 1B and 1C- would make the labeling of the metabolites used a little clearer either in the figure or in the legend- 1b and 1c are not specifically called out in the text and it took me a bit and matching up the HRs to figure out what these represented.

>We now provide full metabolite names plus abbreviations in Figure 1, which will make the figure hopefully clearer.

5. Supplemental Figure 1- the baseline and insulin stimulated pAKT and AKT bands look very different for NogoR and CRELD. This seems unexpected since CRELD is being used for a control for impacts on

insulin signaling? Also big differences in the loading control in the CRELD blot make it a little hard to interpret the changes in pAKT.

>We have now provided alternative blots in **SFig. 8** (previously **SFig. 1**). However, we note that the measurements of the actions of CRELD1 were explored on completely separate cell cultures and therefore that differences in basal p-Akt between these experiments are those exploring NogoR action are not entirely unexpected given inter-culture variability.

6. Table S2- some of the values for Lipidomics ANDIS discovery cohort seem like they are incorrect/were entered incorrectly (average BMI of 60?) diabetes duration of 809 years?

>We apologize for this oversight and have corrected Table S2.

Reviewer #5 (Remarks to the Author):

Using three different cohorts consisting of ~3000 type 2 diabetes patients, the authors conducted multi-omics study of metabolomics, lipidomics and proteomics to identify biomarkers potentially useful for disease prognostics, disease mechanisms, and potential therapeutics to slow diabetes progression. Although the authors validated the results using data from four independent cohorts.

One of the important components of this paper is the statistical methodology for analyzing these complex high dimensional data which have complex data structures, (a) complex correlation structures between and within subjects and across data types, (b) potential missing values, (c) potential nonlinear relationships with covariates, and (d) high dimensionality in the data etc. The section on statistical analysis provides very limited description about the methods used. They provide a list of software packages along with Cox regression, logistic regression and ANOVA. All statistical methods and software make some assumptions regarding the data. For example, ANOVA assumes that the data are (at least asymptotically) normally distributed and more importantly, assumes that all groups have equal variance (homoscedasticity) for each outcome variable. These assumptions have not been verified. Increasingly, researchers are recognizing that metabolomics data are compositional. Are the methods used in this paper account for the compositionality of the data? The Cox regression model makes proportional hazard assumption. Is that valid? Given that the data are being integrated from multiple cohorts, there is a potential for heterogeneity. How is that accounted for? The Benjamini-Hochberg procedure is valid under some conditions. Are those conditions valid in the present analysis?

In summary, the section on Statistical Analysis must be expanded considerably to address various issues raised. Validity of the results of the paper hinge on the appropriateness of the statistical methods.

We apologize for omitting this information. To address your concerns and that of the other reviewers we have now added 1) a flow chart of all the analyses performed and on which cohort in the current study 2) expanded the statistical methods section.

Regarding the compositional nature of the metabolomics. The metabolite data included is not compositional. Metabolites measurements were performed using UHPLC-MS/MS and the levels of the individual metabolites are independent from each other.

Regarding the Cox models. We have verified the proportional hazards assumption of the Cox models using the `cox.zph` function as implemented in the R package `survival`. Heterogeneity of the cohorts was included in the tables (I^2) but there were no signs of heterogeneity across cohorts for the top hits. We have included this in the methods.

Regarding the conditions of the Benjamini Hochberg procedure. The reviewer is correct that Benjamini and Hochberg (1995) have some conditions to perform this procedure one of them being that the tests are independent. In the case of the metabolites and the proteins the measurements are independent of each other. For the lipids, this assumption is possibly violated given that the lipids show a high correlation. However if one would for example use Bonferroni, the same lipids would pass the significance level an alpha of $3.09 \cdot 10^{-4}$ (0.05/162 lipids). For consistency we chose to perform the same multiple testing across the three biomarkers types.

The updated methods (**Page 21, Lines 500-538**)

A Cox proportional hazard model was used to identify molecular risk factors for time to insulin requirement in R (v3.6.0) remotely on each cohort federated node using the `dssCoxph` function in `dsSwissKnife`.²³ Data was log transformed and scaled before analysis. Missing data in the omics were not imputed, but instead individuals were excluded that specific biomarker. The proportionality assumption was assessed using the `cox.zph` function from the survival R package. In each of the three cohorts DCS, GoDARTS and ANDIS, three Cox proportional hazard models were performed:

Model 1: Biomarker, age, sex, BMI, biomarker

Model 2: Biomarker, age, sex, BMI, HDL, C-peptide, biomarker

Model 3: Biomarker, age, sex, BMI, HDL, C-peptide, diabetes duration, glucose-lowering drugs, biomarker

All three models were stratified for HbA1c (strata: < 53 mmol/mol, 53-75 mmol/mol and >75 mmol/mol). The effect of diabetes duration was also investigated on itself, but this did not influence the results, with high correlation between effect sizes before and after adjustment: metabolites 0.96 (95%CI:0.89-0.98), lipids 0.95 (95%CI:0.93-0.96) and proteins 0.91 (95%CI:0.91-0.93). Results from the three cohorts were meta-analyzed using the `metagen` function from the meta R-package. Heterogeneity of across cohorts was assessed using the I² metric. P-values were adjusted for multiple testing using the Benjamini Hochberg procedure. An FDR P-value below 0.05 was considered significant. For the validation in MDC-CC, lipids and proteins identified in the discovery cohorts were tested against incident diabetes using Cox proportional hazard model on a local machine adjusted for age, sex and BMI. In the AGES- Reykjavik cohort, identified proteins were tested against incident and prevalent type 2 diabetes using logistic regression, adjusted for age and sex. In DESIR, logistic regression adjusted for age, sex and BMI was used to test for an association between metabolite levels and prevalent and incident diabetes.

The number of acyl chain length and number of double bonds are important for the direction of effect for TAGs.²⁹ The number of double bonds and the acyl chain length was compared to the hazard ratio observed for TAGs associated with time to insulin initiation.

For the insulin secretion assay and the caspase assay the effect of protein exposure was compared to vehicle using one-way ANOVA. In the animal studies, Student's T-test was used to compare the NogoR group with the control group for body weight, OGTT, insulin tolerance test (ITT), plasma NogoR levels. Differences in pAKT, AKT, IR, pIR between groups were tested using ANOVA and Student's t-test.

Figures and meta-analysis were performed locally with R (v4.0.3). Figures were made using `ggplot2` (v3.3.2). Analysis of cellular and metabolic data were performed using `GraphPad Prism` versions 7.0-9.0 (San Diego, CA, U.S.A.).

The reviewer also refers to correlation between markers. In response to reviewer 3, we have now included a heatmap with the correlation between identified markers.

We have added this finding to the Results (Page 25, Line 597):

Nonetheless, the levels of TAGs were strongly correlated among each other (SFig. 5).

And results (Page 25-26, Lines 613-614):

Levels of testican-1 and HEMK, SMAC, coactosin-like protein were correlated (SFig. 5). Of note, at baseline not all proteins showed a clear upregulation in level in insulin users; the most profound effects were observed for NogoR, IL-18 Ra, ENPP7, HSP 90b (SFig. 6).

REVIEWER COMMENTS

Reviewer #1 (Remarks to the Author):

The manuscript attempts to integrate metabolomics, lipidomics, and proteomics together across multiple cohorts to identify markers of “diabetes progression” that was defined as time to insulin. The major limitation of the study remains, that very limited numbers of analytes were measured across all cohorts and there was no data for time to insulin in replication cohorts for lipidomics and limited data in proteomics (4/11 measured as shown in Fig 3) and their functional studies did not occur in diseased animals, only in those with insulin resistance/risk (i.e. high fat fed or ob/ob mice). They also did not integrate any of the separate omics data together. They did do MR and found 1 metabolite/lipid to be possibly causal PE 18:0;0_18:2:0, and 3 proteins (GDF15, IL-18Ra, and FAS associated with diabetes progression). They pursued functional studies in mice/cell lines with NogoR and IL-18 but I find their conclusions specifically for IL-18Ra to be problematic. They addressed many of my concerns but they did not understand the suggestion of confirming aptamer specificity with GWAS data and again I found their response to my questions about IL-18Ra problematic.

Additional details are provided below:

Specific comments:

Re: Comment #3 response

It is interesting to see that BMI and c-peptide levels are effect modifiers—albeit weak to modestly—of some of these metabolite associations. The wording of this added information, however, is confusing. I would recommend for the interpretation of these results stating: “Of note, for AADA higher levels were observed at baseline for incident insulin users versus non-insulin users, but not for homocitrulline (SFig. 2). AADA also had an interaction with c-peptide levels ($P=0.01$). Homocitrulline did not but did have a modest interaction with BMI ($P=0.03$) which could have masked to some extent, any differences in baseline levels between incident insulin users and non-users.”

Re: Comment #7 response

I was interested in the aptamer-SNP associations via GWAS since SNPs that map to the cognate or coding gene for the protein suggest that the aptamer is measuring the correct protein. For example, what was reported by Sun et. al. in their Nature manuscript (PMID: 29875488). This method, along with confirmation using correlation data with ELISAs of the proteins have been used to confirm the aptamer specificity. Given the publicly available GWAS databases, some attempt could be made to confirm aptamer specificity.

Re: Comment #11 response

While I agree the IL-18Ra could be positively associated with the primary outcome without positively contributing to the disease process, I do not see evidence from this conclusion in the data that is currently presented. While the cohort study association is an inverse association, there was a signal for increased apoptosis in the mouse islet cell study. Data in Fig 7a shows that IL-18 induced NF-kappaB activity is suppressed by differing doses of IL-18Ra, but this is not in a clear linear fashion. Therefore, I find it difficult to conclude that higher IL-18Ra reflect a compensatory mechanism to suppress higher levels of IL-18 activity that leads to islet cell death (whether via interaction with SIGIRR or IL-37, etc.)—which is what I believe the authors are attempting to argue. Also, unclear what data has been presented to support the claim made in line 790-791 in the discussion section. These conclusions should be reconsidered.

Reviewer #2 (Remarks to the Author):

no further comments

Reviewer #3 (Remarks to the Author):

The revised version by Rutter G. et coll. has been overall substantially improved with additional supporting data to explain and support the findings, method and data processing information and adjustment of claims to more specifically fit to the results.

The authors addressed and responded satisfactorily all the points I raised. I recommend the manuscript for publication with minor revision:

- Please provide in the lipidomics method part the descriptor of the commercially available plasma that was used as reference for omic study, lipidomic. It is important for study and methods cross-reference whether the data were collected on NIST reference plasma sample or other commercially available plasma.

- The number of animals used for metabolomic tests should be included in the materials and methods not only in the Figure. It is an essential information for the reliability of the method and study design in general.

-Regarding the observation that the nr of multi-omic markers is low to enrich for pathway analysis, do the authors have an hypothesis why relatively few markers are identified in these cohorts? What perspectives studies based on here presented findings can be done to expedite the mechanistic/pathway analysis?

I think that would be a comment on these aspects in discussion and conclusions would add value to the paper.

Reviewer #4 (Remarks to the Author):

The authors have satisfactorily addressed concerns raised in my prior review.

Reviewer #5 (Remarks to the Author):

Authors have generally addressed my comments. There are some typos. "Student's T" and "Student's t". Are the confidence intervals presented in Figures 3 and 6 adjusted for multiple simultaneous confidence intervals. Given the number of confidence intervals in each figure, it is important to adjust for multiplicity. Not sure if the authors did that. That is important.

REVIEWER COMMENTS

Reviewer #1 (Remarks to the Author):

The manuscript attempts to integrate metabolomics, lipidomics, and proteomics together across multiple cohorts to identify markers of “diabetes progression” that was defined as time to insulin. The major limitation of the study remains, that very limited numbers of analytes were measured across all cohorts and there was no data for time to insulin in replication cohorts for lipidomics and limited data in proteomics (4/11 measured as shown in Fig 3) and their functional studies did not occur in diseased animals, only in those with insulin resistance/risk (i.e. high fat fed or ob/ob mice). They also did not integrate any of the separate omics data together. They did do MR and found 1 metabolite/lipid to be possibly causal PE 18:0;0_18:2:0, and 3 proteins (GDF15, IL-18Ra, and FAS associated with diabetes progression). They pursued functional studies in mice/cell lines with NogoR and IL-18 but I find their conclusions specifically for IL-18Ra to be problematic. They addressed many of my concerns but they did not understand the suggestion of confirming aptamer specificity with GWAS data and again I found their response to my questions about IL-18Ra problematic.

>We are grateful to the referee for re-reading our revised manuscript, and for his/her careful comments.

>We would not wholly agree, however, that the list of analytes was short: an important aspect of the current study is that we have, in the same subjects, measured analytes *across three chemical classes* (162 lipids, 19 small charged molecules and 1195 proteins) and explored interactions between these. To the best of our knowledge this is the first study of this size exploring analytes across three chemical classes in relation to glycemic progression, one of the major challenges in diabetes treatment.

>We would also respectfully challenge the view that the animal models used here were not “diseased”; both the high fat and especially the *db/db* models are well recognized – if imperfect – models of human type 2 diabetes. Whilst the reviewer may have in mind even more profoundly diabetic models (e.g., streptozotocin-treated, non-obese diabetic (NOD) mice, or models based on the deletion of critical transcription factors, e.g. our beta cell selective Pax6 knockout mouse, <https://pubmed.ncbi.nlm.nih.gov/28377501/>), we would argue that the value of these models in the context of the present studies is limited, since they almost inevitably (as required to generate really profound hyperglycemia, and ultimately complications such as kidney disease) have little or no remaining beta cell function: these are essentially models of type 1 diabetes. As such, they preclude meaningful assessment of biomarker action on islet cells similar to subjects with type 2 diabetes, where significant beta cell function usually remains. Of course, these more severe disease models may allow one to examine impacts of a given biomarker on tissues affected by complications (kidney, retina, peripheral nerves, etc). However, since our studies in man used glycemic deterioration - of which damage to these tissues is a consequence but not usually a cause – they would bring little if any new information as to the mechanisms through which these implicated biomarkers might act on glucose homeostasis.

>Indeed, by using two models here (HFD and *db/db*), and a suite of cellular preparations (human and mouse islets, murine hepatocytes and several cell lines) the present studies go substantially further than most cohort-based studies identifying novel biomarkers in human samples.

>We do agree that alternative approaches, such as “MOFA” (Argelaguet, R), which seek to group biomarkers to allow the identification of “factors” may provide enhanced predictive power. Although these analyses lie outside the scope of the present, already complex study, we now insert a statement in the Discussion (Lines 869-871) raising this as a possibility in the future on the present or extended datasets.

Thirdly, the interactions between metabolite classes was not explored in detail, and future studies may generate groups of biomarkers (“factors”) which when considered together collectively provide improved predictions of disease progression.⁸³

Additional details are provided below:

Specific comments:

Re: Comment #3 response

It is interesting to see that BMI and c-peptide levels are effect modifiers—albeit weak to modestly—of some of these metabolite associations. The wording of this added information, however, is confusing. I would recommend for the interpretation of these results stating: “Of note, for AADA higher levels were observed at baseline for incident insulin users versus non-insulin users, but not for homocitrulline (SFig. 2). AADA also had an interaction with c-peptide levels (P=0.01). Homocitrulline did not but did have a modest interaction with BMI (P=0.03) which could have masked to some extent, any differences in baseline levels between incident insulin users and non-users.”

>We agree that this is clearer. We have rephrased according to the reviewer’s suggestion.

Re: Comment #7 response

I was interested in the aptamer-SNP associations via GWAS since SNPs that map to the cognate or coding gene for the protein suggest that the aptamer is measuring the correct protein. For example, what was reported by Sun et. al. in their Nature manuscript (PMID: 29875488). This method, along with confirmation using correlation data with ELISAs of the proteins have been used to confirm the aptamer specificity. Given the publicly available GWAS databases, some attempt could be made to confirm aptamer specificity.

>We apologize that we misinterpreted this suggestion, since the matter raised is quite relevant. Cis-QTLs for the proteins measured on the same platform were collected from Sun et al (Nature, 2018) and Ferkingstad (2021). We show that for at least 6 proteins, highly genome-wide significant SNPs could be identified near the gene of interest (P-value < 1.0x10⁻¹⁵²). For the others more modest, though often nominally significant, signals were observed, which suggests that the specificity of these cannot be confirmed without doubt. Note that the IL18Ra aptamer in Ferkingstad is a different one from that used here, as such QTLs were identified using data from Sun et al 2018.

Seq_Id	ID	Protein	Cis-pQTL	P-value	Source
4374-45_2	SL003869	GDF15/MIC-1	rs1058587	3.8·10 ⁻⁴⁷⁷	Ferkingstad
5105-2_3	SL005208	Nogo receptor	rs75766	4.1·10 ⁻²⁰⁴	Ferkingstad
3446-7_2	SL004152	IL18 Ra	rs1420106	1.1·10 ⁻²⁷³	Sun
7628-40_3	SL012774	CRELD1	rs4234585	1.3·10 ⁻²⁶⁴⁰	Ferkingstad
4435-66_2	SL009045	ENPP7	rs11871061	7.9·10 ⁻²⁹⁵³	Ferkingstad
5392-73_2	SL002731	Fas, soluble	rs7911226	1.0·10 ⁻¹⁵²	Ferkingstad
5467-15_3	SL000454	HSP-90B	rs190077456	7.80·10 ⁻⁶	Ferkingstad
3122-6_2	SL003733	SMAC	rs111553874	1.10·10 ⁻⁴	Ferkingstad
4905-63_1	SL004814	Coactosin-like protein	rs173777	3.27·10 ⁻⁴	Ferkingstad
5490-53_3	SL010384	Testican-1	rs11744278	7.5·10 ⁻⁴	Ferkingstad
11096-57_3	SL018921	HEMK2	rs1470967471	0.003155	Ferkingstad

>SNPs included in the table are only those with MAF \geq 1% and 1Mb from transcription start or end site. The study from Ferkingstad et al is a GWAS based on 35,559 individuals.

>We have added this information to the method section (Lines 259-264):

To verify the specificity of the aptamers, we confirmed the presence of a cis-QTL associated with top identified proteins described previously as method to confirm aptamer specificity.^{21,22} SNPs were considered when the minor allele frequency was \geq 1% and 1Mb from transcription start or end site. The primary set used was that of Ferkingstad et al.²² given the large population, but when aptamers were not included in that particular set, the Sun et al. study was used.²¹

>And to the results (Lines 632-635):

*Finally, aptamer specificity of the top proteins was verified by confirming the presence of a cis-QTL. Out of the eleven proteins, six proteins showed a strong QTL at the genome-wide significance level: GDF15, NogoR, IL18 Ra, CRELD1, ENPP7 and Fas (**Table S8**).*

Re: Comment #11 response

While I agree the IL-18Ra could be positively associated with the primary outcome without positively contributing to the disease process, I do not see evidence from this conclusion in the data that is currently presented. While the cohort study association is an inverse association, there was a signal for increased apoptosis in the mouse islet cell study. Data in Fig 7a shows that IL-18 induced NF-kappaB activity is suppressed by differing doses of IL-18Ra, but this is not in a clear linear fashion. Therefore, I find it difficult to conclude that higher IL-18Ra reflect a compensatory mechanism to suppress higher levels of IL-18 activity that leads to islet cell death (whether via interaction with SIGIRR or IL-37, etc.)—which is what I believe the authors are attempting to argue. Also, unclear what data has been presented to support the claim made in line 790-791 in the discussion section. These conclusions should be reconsidered.

> We agree and have modified this discussion slightly, as suggested, inserting the following statement (Lines 828-832):

Nevertheless, the contrasting actions of IL-18Ra (impaired IL-18 signaling in the HEK cell model of receptor overexpression, but increased apoptosis in primary islets) suggest that further studies, possibly involving tissues-specific inactivation of IL-18Ra or other potential receptors (see above), will be needed to fully elucidate the (patho)physiological roles of circulating IL-18Ra.

RE (previously) line 790-791 in the discussion section:

This is now clarified: we referred to the study of Reznikov et al (ref 71):

We confirmed earlier studies⁷¹ which demonstrated that an IL-18Ra:Fc fusion fragment inhibits the pro-inflammatory action of IL-18. However, and in distinction to these earlier studies⁷¹, measuring interferon-gamma (IFN-gamma) production from mononuclear cells, the actions of IL-18Ra, as examined in the present study, did not depend upon the additional presence of IL-18Rbeta.

Reviewer #2 (Remarks to the Author):

no further comments

> We thank the reviewer for his/her feedback.

Reviewer #3 (Remarks to the Author):

The revised version by Rutter G. et coll. has been overall substantially improved with additional supporting data to explain and support the findings, method and data processing information and adjustment of claims to more specifically fit to the results. The authors addressed and responded satisfactorily all the points I raised. I recommend the manuscript for publication with minor revision:

> We thank the reviewer for the thorough reading and feedback.

- Please provide in the lipidomics method part the descriptor of the commercially available plasma that was used as reference for omic study, lipidomic. It is important for study and methods cross-reference whether the data were collected on NIST reference plasma sample or other commercially available plasma.

> We haven't used any commercial reference plasma. Instead, we used an internally created reference based on plasma of subjects without unusual supplements or medication purchased from the Deutsches Rotes Kreuz, Kreisverband Dresden e.V., that we run alongside study samples in order to control for technical variation and overall analytical quality of the run. This has now been further clarified in the text (Lines 243-244):

The reference samples are replicates from a pool of plasma purchased from the Deutsches Rotes Kreuz, Kreisverband Dresden e.V..

- The nr of animals used for metabolomic tests should be included in the materials and methods not only in the Figure. It is an essential information for the reliability of the method and study design in general.

> This has been added to the Methods.

-Regarding the observation that the nr of multi-omic markers is low to enrich for pathway analysis, do the authors have an hypothesis why relatively few markers are identified in these cohorts? What perspectives studies based on here presented findings can be done to expedite the mechanistic/pathway analysis?

I think that would be a comment on these aspects in discussion and conclusions would add value to the paper.

> One of the reasons that we find relatively few markers in the current study may be the complexity of the endpoint, which is influenced by many factors, including individual choices, therapy adherence, local prescribing protocols etc. Nonetheless the metabolites, lipids and proteins that we identify are those that have a sufficiently strong signal to overcome all these influencing factors and may thus serve as robust biomarkers of glycemic progression. Larger discovery and validation studies would likely identify additional biomarkers but were beyond the scope of the current studies. We provide examples of follow-up work one can do to get more insight into the functional role of proteins such as NogoR and IL18Ra. Similar studies could be performed for the other biomarkers to elucidate potential mechanisms of action. Nonetheless, have added a statement regarding the integration of markers to the discussion in Lines 869-871:

Thirdly, the interactions between metabolite classes was not explored in detail, and future studies may generate groups of biomarkers ("factors") which when considered together collectively provide improved predictions of disease progression.⁸³

Reviewer #4 (Remarks to the Author):

The authors have satisfactorily addressed concerns raised in my prior review.

> We thank the reviewer for critical evaluation of our manuscript.

Reviewer #5 (Remarks to the Author):

Authors have generally addressed my comments. There are some typos. "Student's T" and "Student's t".

> Modified.

Are the confidence intervals presented in Figures 3 and 6 adjusted for multiple simultaneous confidence intervals. Given the number of confidence intervals in each figure, it is important to adjust for multiplicity. Not sure if the authors did that. That is important.

> We thank the reviewer for this additional comment. We recognize the importance of this suggestion, especially for the confidence intervals presented in Figure 1-3 and Supplemental Tables 5-7, where we adjusted the P-values for multiple testing using the BH-procedure. As such, we have modified confidence intervals of the meta-analysis in these figures and added the adjusted confidence intervals to the respective tables. Note that the individual studies (DCS, GoDARTS and ANDIS) shown are there just for illustrative purposes and to provide insight into the heterogeneity of the studies used in the random effects meta-analysis. Moreover, we have updated the confidence intervals in the manuscript text itself. We have added this to the methods (529-534) and figures and Suppl. Tables:

*Reported confidence intervals were adjusted for multiple simultaneous confidence intervals. For this, instead of showing the 95% confidence intervals, the interval was based on the number of tests and significant hits at $FDR < 0.05$. Specifically, the adjusted confidence interval to be used was calculated as $1 - q * R / m$, where q is the level at which the FDR is controlled (0.05), R the number of significant tests at 5% FDR and m the total number of tests performed.³¹*

>The reviewer also refers to Figure 6, but note that this figure is based on two mouse strains and independent experiments. Within OGTT and ITT it is not required to adjust for multiplicity and as such we prefer to keep the current means+SEMs. However, we have added a sentence in the legend of the figure to alert the readers that the CIs were not adjusted for multiple simultaneous CIs:

Data are represented as the mean \pm SEM unadjusted for multiple simultaneous CIs.

Other minor change

We have also slightly modified a paragraph in the Discussion (lines 777-783), providing a further relevant reference:

The mechanisms involved in the latter action are unclear given very low levels of expression of the receptors NogoR (cell attached), p75 and OMGP in islets⁶² (BioGPS.org) though the disruption of an interaction between the NogoR/p75 complex and the co-receptor Lingo-1⁶³, also expressed in islets, by soluble NogoR, is conceivable. Since RTN4, encoding NogoA, NogoB and NogoC via alternative splicing⁶⁴, as well as NgBR, are also expressed in islet endocrine cells

(<https://huisinglab.com/data/index.html>), circulating NogoR might also disrupt NogoB binding to NgBR on β cells.

REVIEWERS' COMMENTS

Reviewer #1 (Remarks to the Author):

I am satisfied with the responsiveness of the authors. I still have some minor concerns regarding aptamer specificity and might modulate their edits to lines 259-264 slightly:

Finally, we examined aptamer specificity of the top proteins by testing for the presence of a cis-QTL. Out of the eleven proteins, six proteins showed a strong QTL at the genome-wide significance level: GDF15, NogoR, IL18 Ra, CRELD1, ENPP7 and Fas (Table S8).

The point is...not all were verified.

Reviewer #5 (Remarks to the Author):

I have no further comments. The authors have addressed all my comments.

REVIEWER COMMENTS

Reviewer #1 (Remarks to the Author):

I am satisfied with the responsiveness of the authors. I still have some minor concerns regarding aptamer specificity and might modulate their edits to lines 259-264 slightly:

Finally, we examined aptamer specificity of the top proteins by testing for the presence of a cis-QTL. Out of the eleven proteins, six proteins showed a strong QTL at the genome-wide significance level: GDF15, NogoR, IL18 Ra, CRELD1, ENPP7 and Fas (Table S8).

The point is...not all were verified.

We thank the reviewer for his/her comments

We have rephrased this sentence:

“Finally, we examined aptamer specificity of the top proteins by testing for the presence of a cis-QTL. Out of the eleven proteins, six proteins showed a strong QTL at the genome-wide significance level: GDF15, NogoR, IL18 Ra, CRELD1, ENPP7 and Fas (Table S8).”

With

Out of the eleven proteins, specificity of six proteins was verified on the basis of a cis-pQTL: GDF15, NogoR, IL18 Ra, CRELD1, ENPP7 and Fas (Table S4).